# CodeInsightBench: A Benchmark for Advanced Code Understanding and Comparison in Large Language Models

## Abstract

While Large Language Models (LLMs) have demonstrated significant progress in coding tasks, their capabilities in nuanced code analysis and deep comprehension remain insufficiently explored. To address this gap, we introduce **CodeInsight-Bench**, a comprehensive multilingual benchmark designed to systematically evaluate advanced code reasoning. Built upon real-world Codeforces data, it employs both multiple-choice and open-ended questions to assess three core tasks: **Semantic Code Judgment**, **Debugging Path Tracking**, and **Code Efficiency Comparison**. We conduct extensive evaluations on **22** state-of-the-art LLMs (11 closed-source, 11 open-source), revealing critical insights into their strengths and limitations. Our results reveal substantial performance gaps across tasks, with closed-source models generally outperforming open-source counterparts. Besides, models fail primarily on large-scale code transformations, indicating fundamental limitations in understanding code evolution logic. Additionally, the results indicate distinct programming language preferences in code efficiency comparison, and show that multiple sampling substantially improves semantic code judgment performance, with Pass@3 achieving 92.71% accuracy compared to 60.57% at Pass@1. By providing a comprehensive and systematic evaluation methodology, CodeInsightBench enables deeper understanding of LLM capabilities in sophisticated code comprehension tasks.

## 1 Introduction

Large Language Models (LLMs) have become increasingly influential in programming, offering significant advancements across multiple domains. In particular, they have made notable contributions in areas such as code generation (Nijkamp et al., 2023; Dong et al., 2024), debugging (Wei et al., 2023; Zhong et al., 2024), and code reasoning (Liu et al., 2024a), revolutionizing the way software development tasks are approached. Tools such as GitHub Copilot (GitHub, 2021) and Cursor (Cursor, 2023) have demonstrated impressive capabilities in automating code generation from natural language descriptions, helped developers streamline the coding process and reduce manual effort. Recent advances in LLMs have enabled them to analyze code logic, predict bugs, and offer optimization suggestions. However, these approaches mainly focused on surface-level tasks, with limited attention given to evaluating their deeper understanding of code, such as the ability to reason about code structure, track changes over time, and assess the solution efficiency (Nam et al., 2024).

Existing studies on code reasoning have largely focused on evaluating code generation accuracy (Liu et al., 2023) and efficiency (Dong et al., 2025), while underemphasizing more complex aspects such as code understanding and comparison. As shown in Table 1, popular code evaluation benchmarks suffer from key limitations: focus on isolated code snippets rather than code relationships, reliance on open-ended QA or biased MCQ formats, and use of publicly available datasets prone to data leakage. This gap highlights the need for a systematic benchmark to assess LLMs' deeper insight into code, moving beyond code generation toward comprehensive evaluation of their comprehension and reasoning capabilities. The ability to answer questions about code represents a natural extension of LLMs' role in programming, requiring models to exhibit profound understanding of code semantics, debugging processes, and optimization strategies.

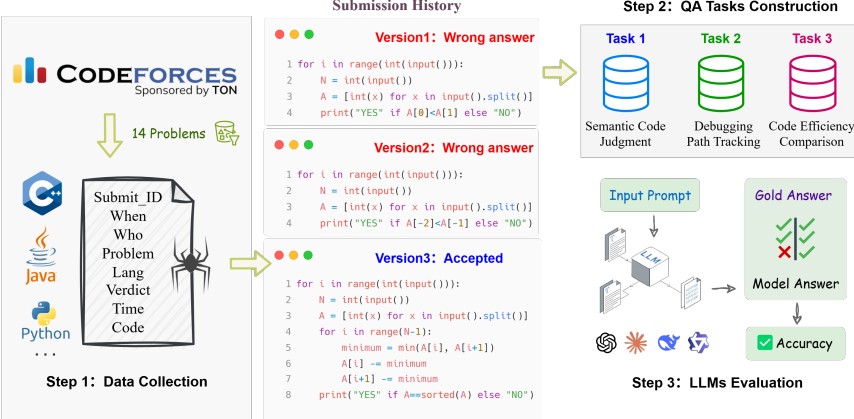

Figure 1: **CodeInsightBench Overview.** The benchmark is constructed by first collecting metadata from Codeforces, followed by the creation of Q&A tasks that focus on key aspects of code understanding, such as semantic judgment, debugging, and efficiency. The final step involves evaluating LLMs by comparing their outputs to a gold standard, enabling the assessment of their accuracy and comprehension of code.

To facilitate comprehensive evaluation of code comprehension and cross-code analysis, we present CodeInsightBench, a novel benchmark that challenges large language models with complex reasoning tasks requiring deep code understanding and comparative analysis. As shown in Figure1, we systematically construct CodeInsightBench using data from Codeforces (Codeforces, 2025), leveraging actual programming contest submissions to create authentic question-answer pairs derived from verified code implementations. Through rigorous rule-based filtering and manual verification, we curate 14,035 high-quality pairs spanning both open-ended and multiple-choice formats. This methodology ensures meaningful evaluation that reflects genuine coding challenges, moving beyond synthetic annotations toward assessment frameworks that enable more intuitive, human-like programming interactions. Our benchmark evaluates deep code understanding through three strategically designed tasks: **Semantic Code Judgment** for semantic comprehension, **Debugging Path Tracking** for logical execution tracing, and **Code Efficiency Comparison** for algorithmic performance analysis. Unlike traditional benchmarks focused on syntactic correctness, these tasks demand profound understanding of code semantics and programming logic, providing comprehensive assessment of LLMs' capacity to handle authentic challenges in debugging, optimization, and code reasoning. Our contributions are summarized as follows:

- We introduce CodeInsightBench, a comprehensive multi-language benchmark featuring 14,035 Q&A pairs derived from real-world code submissions. While existing benchmarks primarily assess single-function analysis and code generation capabilities, our benchmark fills a critical gap by evaluating LLMs' ability to perform cross-code comparative reasoning across three challenging tasks.
- Through extensive evaluation of 22 state-of-the-art models (11 closed-source and 11 open-source), we reveal significant performance gaps in cross-code understanding and uncover intriguing patterns: models excel at surface-level syntax comparison but struggle with deeper code reasoning. Notably, some reasoning models like OpenAI o-series exhibit a confidence miscalibration phenomenon, where high accuracy comes at the cost of trustworthy uncertainty estimation.

## 2 RELATED WORK

**Code Benchmark.** Previous benchmarks, such as HumanEval (Chen et al., 2021), MBPP (Austin et al., 2021), and DS-1000 (Lai et al., 2023), have primarily focused on evaluating code generation capabilities but lacked in-depth assessments of models' broader code-related abilities. As LLM coding capabilities have expanded, the scope of benchmarks has diversified to include additional tasks, such as code efficiency evaluation (e.g., EFFIBENCH (Huang et al., 2024), Mercury (Du et al., 2024)) and code repair (e.g., MdEval (Liu et al., 2024b), DebugBench (Tian et al., 2024), Feed-

Table 1: The comparison between popular code evaluation benchmarks and CodeInsightBench.

| Benchmark | Task Type | | | | Data Properties | | |
| --- | --- | --- | --- | --- | --- | --- | --- |
| | Code Understanding | Code Comparison | Code Q&A Open-end | MCQ | Multilingual | Real-world | Test Size |
| EFFIBENCH | ✗ | ✗ | ✗ | ✗ | ✗ | ✓ | 1,000 |
| MdEval | ✓ | ✓ | ✗ | ✓ | ✓ | ✓ | 3,897 |
| DebugBench | ✓ | ✓ | ✓ | ✗ | ✓ | ✗ | 4,253 |
| FeedbackEval | ✓ | ✗ | ✗ | ✗ | ✗ | ✓ | 3,736 * |
| Infibench | ✓ | ✓ | ✓ | ✗ | ✓ | ✓ | 234 |
| CRQBench | ✓ | ✓ | ✓ | ✗ | ✗ | ✓ | 100 |
| CoCo-Bench | ✓ | ✗ | ✗ | ✗ | ✓ | ✓ | 705 |
| CodeXGLUE | ✓ | ✓ | ✗ | ✗ | ✓ | ✓ | 2.75M * |
| CodeCriticBench | ✗ | ✗ | ✓ | ✗ | ✓ | ✗ | 4,300 |
| LiveCodeBench | ✓ | ✗ | ✓ | ✗ | ✗ | ✓ | 1,432 |
| CodeJudgeBench | ✓ | ✓ | ✗ | ✓ | ✗ | ✗ | 4,260 |
| CodeMMLU | ✓ | ✓ | ✗ | ✓ | ✓ | ✓ | 19,912 * |
| **CodeInsightBench** | ✓ | ✓ | ✓ | ✓ | ✓ | ✓ | **14,035** |

*\* Indicates that the data is sourced from publicly available existing datasets.*

backEval (Dai et al., 2025)). Simultaneously, the evaluation landscape has evolved to incorporate novel methodological approaches, with several benchmarks adopting multi-task approaches to comprehensively evaluate code comprehension from multiple perspectives. For instance, CoCo-Bench (Yin et al., 2025) and CodeXGLUE (Lu et al., 2021) assess models across various tasks, including code inference, translation, repair, and summarization.

**QA-based Code Evaluation.** Benchmarks like Infibench (Li et al., 2024) and CRQBench (Dinella et al., 2024) focus on open-ended code-related question answering. CodeCriticBench (Zhang et al., 2025) and LiveCodeBench (Jain et al., 2024) utilize open-ended code QA to evaluate models on diverse code-related tasks. These approaches typically source question-answer pairs from open platforms or manual annotations, but the quality can vary significantly, and high annotation costs often result in smaller datasets with limited coverage. Alternatively, multiple-choice benchmarks like CodeMMLU (Manh et al., 2024) and CodeJudgeBench (Jiang et al., 2025) provide more stable evaluation with reduced annotation overhead. CodeMMLU specifically derives its data from existing public datasets and programming resources. However, CodeMMLU's reliance on existing public datasets and programming resources raises concerns about data leakage, as these sources may have been included in LLM training data. Moreover, MCQ approaches face inherent limitations: they require carefully crafted distractors and may introduce bias when generating candidate answers through automated processes or LLM assistance, potentially limiting their effectiveness in assessing natural reasoning patterns and authentic problem-solving approaches.

## 3 CODEINSIGHTBENCH

In this section, we introduce CodeInsightBench, a novel benchmark designed to assess LLMs' ability to understand programming contest challenges in depth. The benchmark uses a code question-answering approach with three evaluation tasks, employing both multiple-choice questions and open-ended QA pairs derived from competitive programming solutions. We detail the data collection, task formulation, and evaluation metrics used to assess model performance.

### 3.1 DATA CURATION

CodeInsightBench utilizes data from Codeforces(Codeforces, 2025), a widely-recognized competitive programming platform, including coding challenges and authentic user submissions. The raw dataset consists of 73,895 submission records for 14 medium-difficulty problems, with ratings ranging from 1000 to 1600. These submissions come from 6,703 unique users and span the period from December 5, 2024, to April 28, 2025. The recency of the dataset ensures that no data leakage into the training data of the evaluated models has occurred. Specific details about the 14 problems can be found in Appendix C.1.

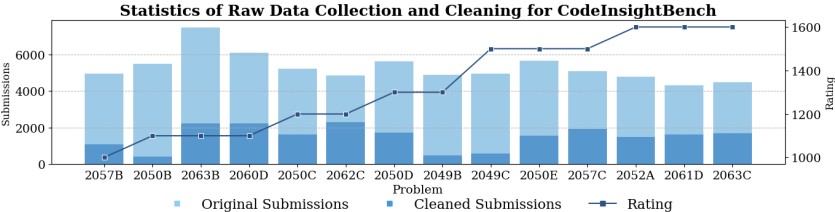

Figure 2: Dataset statistics for CodeInsightBench. The figure shows the problem rating distribution and submission counts in both the original raw dataset and the curated dataset used for question-answer pair construction.

Given the inherent noise and inconsistencies in raw user submission data, comprehensive data cleaning procedures were essential to ensure dataset quality and reliability. We implement systematic filtering criteria to eliminate problematic submissions and maintain data integrity. Specifically, we excluded submissions from users who consecutively submitted solutions in multiple programming languages for the same problem to reduce redundancy and maintain consistency. We also removed submission records that did not result in a final verdict of accepted(AC). To ensure effective code debugging trajectories, we excluded submissions with more than five attempts to focus on meaningful debugging processes while avoiding overly complex cases. Following these rigorous cleaning procedures and manual validation, we retained 20,856 high-quality submissions that form the foundation of our benchmark dataset. Figure 2 presents the statistical overview of the curated dataset utilized for question-answer pair construction. Detailed information about the constructed Q&A datasets for each task is provided in the Appendix C.2.

## 3.2 TASK CONSTRUCTION

As illustrated in Figure 3, CodeInsightBench offers multi-dimensional insights into LLMs' code understanding and comparison capabilities through three core tasks—Semantic Code Judgment, Debugging Path Tracking, and Code Efficiency Comparison. The prompt template used in these three tasks are detailed in Appendix D.4.1.

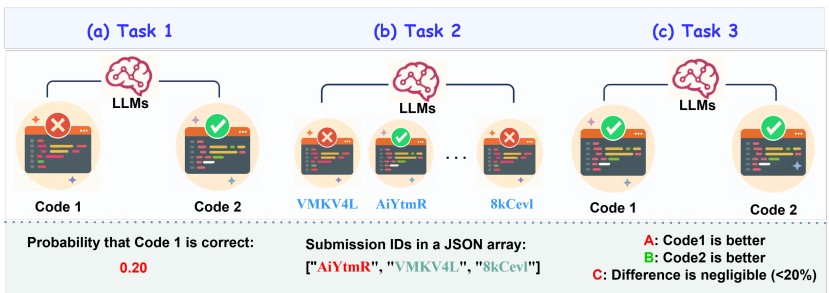

Figure 3: Overview of CodeInsightBench's three tasks: (a) Semantic Code Judgment; (b) Debugging Path Tracking; (c) Code Efficiency Comparison. Each task presents one or more contest solutions as input and requires the model to output respectively a correctness judgment score, a step-by-step execution trace, or a relative performance decision.

**Task 1: Semantic Code Judgment.** Unlike traditional code correctness evaluation that targets obvious syntactic or logical errors, our Semantic Code Judgment task evaluates models' ability to distinguish between semantically similar code submissions from the same user, requiring reasoning about how subtle modifications can fundamentally alter program behavior. In this task, the model is given pairs of code submissions where one is correct, and the other contains errors resulting in verdicts such as *Wrong answer*, *Compilation error*, *Runtime error*, *Time limit exceeded*, or *Memory limit exceeded*. The model's objective is to identify the correct submission. To avoid static binary outputs, we introduce a open-ended probabilistic evaluation approach that requires the model to output the probability that the first code submission receives an *Accept* verdict.

**Task 2: Debugging Path Tracking.** Real-world debugging involves gradual refinement through iterative code modifications, making the understanding of these evolutionary patterns crucial for evaluating models' code comprehension capabilities. The Debugging Path Tracking task evaluates the model's ability to understand code evolution patterns by presenting sequences of submissions from the same user for a specific problem, each reflecting iterative attempts to reach an accepted solution (AC). Models must identify the underlying debugging trajectory by recognizing syntactic and semantic variations that indicate logical progression patterns. To prevent position-based bias, all submissions are randomly shuffled and anonymized with modified identifiers. Models are requested to predict submission ID sequences, which are evaluated against ground truth chronological orderings to measure proficiency in code evolution reasoning and comparative analysis.

**Task 3: Code Efficiency Comparison.** While correctness verification remains fundamental in code evaluation, assessing algorithmic efficiency represents a more sophisticated dimension of code understanding that demands deeper analytical capabilities from large language models. The Code Efficiency Comparison task evaluates a model's ability to compare the relative efficiency of two verified correct code submissions using the same programming language. The objective is to determine which implementation demonstrates superior efficiency based on both theoretical time complexity analysis and empirical execution performance. This is an MCQ task where models choose between different scenarios. To mitigate the impact of minor performance variations inherent in different execution environments and input configurations, we establish an equivalence threshold whereby submissions are considered equally efficient when their execution time difference is less than 20% of their average execution time. This threshold effectively filters measurement noise while preserving meaningful efficiency distinctions.

## 3.3 DATASET ANALYSIS

**Task 1: Semantic Code Judgment.** Task 1 contains 1,400 question-answer pairs across 14 problems, with 100 pairs per problem. To avoid positional bias, correct and incorrect submissions are randomly assigned to code positions, resulting in a nearly balanced distribution. The problems span diverse algorithmic categories including dynamic programming, graph algorithms, and mathematical computations, ensuring comprehensive coverage of programming concepts. Detailed analysis of the character-level edit distance distribution is provided in Appendix C.3, revealing that our dataset captures both subtle semantic differences and more substantial code variations.

**Task 2: Debugging Path Tracking.** Task 2 consists of 3,314 debugging paths from individual users addressing a single problem. The distribution includes 1,869 three-submission paths, 964 four-submission paths, and 481 five-submission paths, reflecting typical debugging cycles in real-world development, where shorter iterations are more common. Each path represents an authentic debugging trace, capturing the iterative refinement process programmers undergo when solving complex problems. The paths encompass various error types including logical errors, edge case handling, and algorithm optimization, providing comprehensive coverage of debugging scenarios.

**Task 3: Code Efficiency Comparison.** Task 3 features 9,321 code pairs across three programming languages: 1,564 in C++17, 1,743 in Java, and 6,014 in Python. The label distribution is balanced, with approximately 35.2% of pairs marked as equally efficient, 32.9% favoring the Code 1, and 31.9% favoring the Code 2. The efficiency comparisons are determined through actual runtime measurements on identical test cases, which serves as a practical proxy for algorithmic efficiency since both codes solve the same problem under identical constraints.

## 3.4 EVALUATION CRITERIA

To evaluate the performance of Large Language Models across the three tasks in CodeInsightBench, we employ four main evaluation metrics: (1) **Accuracy (Acc.)** measures the proportion of correctly predicted instances among all evaluated examples. (2) **Cross-Entropy (CE)** evaluates how well the predicted probability distributions align with the ground-truth labels, with lower values indicating better performance. (3) **Pairwise Accuracy (Pair Acc.)** assesses the model's ability to maintain correct relative rankings between pairs of items by comparing the predicted order with the ground-truth ranking. (4) **Pass@$k$** determines whether the correct answer appears within the top-$k$ predictions, which is particularly valuable for tasks allowing multiple valid outputs. Detailed mathematical definitions and formulations for all evaluation metrics are provided in Appendix C.4.

Table 2: Results of different models. The evaluation covers 11 closed-source and 11 open-source models across 3 tasks in CodeInsightBench.

| Type | Model Name | Size | Task 1 | | Task 2 | | Task 3 |
|------|-----------|------|--------|------|--------|----------|--------|
| | | | Acc.↑ | CE↓ | Acc.↑ | Pair Acc.↑ | Acc.↑ |
| Close-sourced | GPT-5 | - | **93.43** | **0.2125** | **61.16** | **83.01** | **45.50** |
| | O4-mini | - | 86.93 | 1.5695 | 45.62 | 73.61 | 43.73 |
| | Claude-Opus-4-1 | - | 75.93 | 0.5873 | 54.83 | 79.63 | 38.65 |
| | O1-mini | - | 68.14 | 3.5528 | 28.09 | 62.74 | 41.81 |
| | Gemini-2.0-Flash-Exp | - | 54.14 | 2.4422 | 33.43 | 66.80 | 35.09 |
| | GPT-4o | - | 63.00 | 0.7632 | 37.72 | 68.74 | 37.61 |
| | GPT-4.1-Mini | - | 71.07 | 1.6420 | 40.13 | 70.33 | 38.86 |
| | GPT-3.5-Turbo | - | 52.64 | 2.9725 | 15.99 | 52.06 | 33.04 |
| | GLM-4-Plus | - | 59.86 | 0.8374 | 23.78 | 59.18 | 36.19 |
| | Claude-3-7-Sonnet | - | 67.43 | 2.5177 | 44.99 | 73.60 | 36.78 |
| | Claude-3-5-Haiku | - | 52.57 | 0.8202 | 21.54 | 57.10 | 35.08 |
| Open-sourced | Qwen3-8B | 8B | 62.21 | 3.1812 | 34.13 | 66.91 | 34.33 |
| | Qwen3-14B | 14B | **62.86** | 3.1017 | 33.22 | 67.00 | 34.29 |
| | Qwen2.5-Coder-7B-Instruct | 7B | 51.93 | 3.8621 | 16.02 | 54.49 | 34.42 |
| | Qwen2.5-7B-Instruct-1M | 7B | 50.00 | **1.0060** | 14.03 | 51.99 | 33.27 |
| | Qwen2.5-7B-Instruct | 7B | 52.64 | 1.4775 | 14.21 | 52.84 | 34.33 |
| | Qwen2.5-14B-Instruct-1M | 14B | 60.21 | 1.0278 | 24.56 | 61.51 | 37.08 |
| | Phi-4 | 14B | 56.86 | 3.4042 | 21.12 | 58.94 | 33.44 |
| | Llama-3.1-8B-Instruct | 8B | 52.36 | 3.5515 | 11.29 | 54.03 | 32.51 |
| | Gemma-2-9B-It | 9B | 55.36 | 4.1816 | 14.24 | 52.73 | 33.27 |
| | Deepseek-V3 | 671B | 59.57 | 1.2582 | **43.39** | **72.30** | **39.20** |
| | DeepSeek-V2 | 236B | 60.36 | 1.2935 | 42.97 | 71.86 | 38.91 |

# 4 EXPERIMENTS

## 4.1 EXPERIMENTAL SETUP

We conduct a comprehensive evaluation of 22 popular LLMs from 9 different organizations, including 11 closed-source and 11 open-source models. Detailed parameter settings and model configurations in our experiments are provided in Appendix D.1.

## 4.2 MAIN RESULTS

The results in Table 2 and Figure 4 demonstrate substantial performance differences among models, highlighting the diagnostic value of CodeInsightBench. Closed-source models consistently outperform their open-source counterparts across all evaluation dimensions, with GPT-5 leading by substantial margins, achieving 93.43% accuracy on Task 1 compared to 62.86% for the top open-source model. Beyond that, we find that o1-mini and o4-mini exhibit a confidence miscalibration phenomenon in Task 1, where high accuracy comes at the cost of trustworthy uncertainty estimation. Particularly noteworthy is that o4-mini surpasses Claude-Opus-4-1 in accuracy yet exhibits nearly 3× worse calibration, suggesting that current reasoning model development may prioritize correctness over reliability. These performance patterns are further visualized in Appendix D.2.

**Task Complexity Hierarchy Reveals Fundamental Gaps in Code Understanding.** Task performance analysis reveals a clear difficulty hierarchy across the benchmark. Task 1 proves most accessible to current models, with top performers like GPT-5 achieving 93.43% accuracy, indicating that static code comparison aligns well with LLM capabilities. Task 2 presents moderate difficulty, where even the best model reaches 61.16% accuracy, reflecting the challenges of reasoning through execution traces. The model performance on Task 2 also varies across different debug path lengths, as detailed in Appendix E.1. Most notably, Task 3 emerges as the most challenging task, with all models struggling significantly—the highest accuracy peaks at only 45.50%, demonstrating that algorithmic complexity analysis and runtime behavior understanding remain substantial hurdles for current language models.

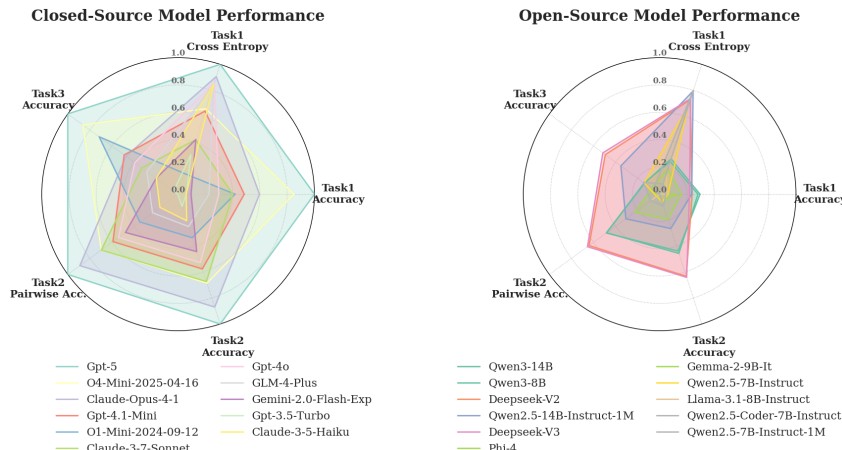

Figure 4: Radar chart comparing the performance of closed-source and open-source models across three tasks in CodeInsightBench. Task 1 evaluates Accuracy and Cross-Entropy, Task 2 assesses Accuracy and Pairwise Accuracy, and Task 3 focuses on Accuracy. The chart highlights that closed-source models generally perform better across tasks, while open-source models demonstrate specialized strengths in certain areas.

**Complex Code Transformations Expose Fundamental Limitations in Semantic Reasoning.** Our analysis reveals that current LLMs systematically fail when reasoning about substantial code evolution, indicating fundamental limitations in semantic understanding beyond syntactic pattern recognition. Table 3 presents collective model failures in the Debugging Path Tracking task, comparing cases where all models fail against those where at least 80% of models answer correctly (similarity calculation detailed in Appendix E.3). These universally challenging scenarios typically involve substantially more complex code structures. By task category, models perform relatively well on Greedy (all wrong 16.47%, 80%+ correct 20.37%) and Sortings (all wrong 6.60%, 80%+ correct 9.35%), but struggle on Bitmasks (all wrong 5.38%, 80%+ correct 3.18%) and Graph matching (all wrong 1.28%, 80%+ correct 0.19%). These findings highlight a critical weakness: models excel at detecting incremental changes but cannot comprehend the broader logical intent underlying longer, more complex, or logic-specific code transformations, indicating a systematic architectural limitation rather than model-specific weaknesses. Developing models capable of learning semantic difference representations between code versions thus represents a key research direction for deeper understanding of how code modifications affect program behavior and logic.

Table 3: Comparative Analysis of Universally Failed vs. High-Agreement Correct Cases in Task 2

| Metric | Universally Incorrect Samples | High-Agreement Correct Samples (≥80%) |
|---|---|---|
| Case Count | 560 | 167 |
| Submissions Count | 4.10 | 3.11 |
| Avg. Similarity | 0.760 | 0.823 |
| Min. Similarity | 0.619 | 0.749 |
| Avg. Diff Lines | 40.08 | 25.71 |
| Avg. Token | 561.65 | 362.97 |

**Poor Algorithmic Reasoning Performance Across All Programming Languages.** To examine language-specific effects on algorithmic reasoning, we evaluate LLM performance across different programming languages in the Code Efficiency Comparison task, with results presented in Figure 5. The evaluation reveals a fundamental challenge: no model achieves satisfactory performance in time complexity assessment, with most accuracy scores falling below 50% regardless of the programming language used. Notably, performance on Python is generally slightly lower than on Java and C++17, and different models exhibit distinct performance patterns across languages. This indicates that LLM performance on algorithmic reasoning and time complexity evaluation depends not only on fundamental reasoning capabilities but also on language syntax, conventions, and training

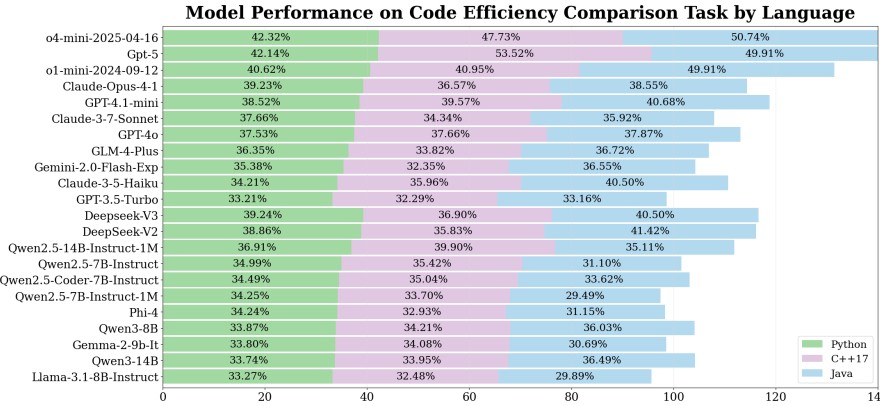

Figure 5: Performance of Models on Time Complexity Assessment Across Different Programming Languages. The stacked bar chart displays the performance of various models on the time complexity assessment task, segmented by programming language (Python, C++17, and Java). Models are compared based on their accuracy across different programming languages, with varying performance trends observed between languages.

data distribution. To further understand model behavior, we conducted comprehensive bias analysis examining overall prediction tendencies, with detailed results presented in Appendix E.2.

### 4.3 MODEL ROBUSTNESS ASSESSMENT

To evaluate whether models can achieve improved performance through multiple sampling attempts, we conduct a Pass@$k$ analysis using GPT-4-Turbo on Task 1 and Task 3. Task 2 is excluded from this analysis due to its complex multi-step path output format, which requires specialized evaluation metrics beyond simple accuracy measures. We evaluate performance with k=1, 2, and 3 attempts, generating independent samples for each test instance. Table 4 presents the Pass@$k$ results across different programming languages, revealing how sampling strategies affect model performance in code understanding and code comparing tasks.

For Task 1, GPT-4-Turbo demonstrates substantial improvement through multiple sampling: overall accuracy increases from 0.6057 at Pass@1 to 0.7493 at Pass@2 and reaches 0.9271 at Pass@3. Java shows the strongest performance at Pass@1 (0.6765) and Pass@2 (0.8235), while C++17 achieves the highest Pass@3 accuracy (0.9442). All languages demonstrate significant improvements with multiple sampling attempts, with Pass@3 accuracy exceeding 0.88 across all tested languages.

For Task 3, the baseline performance is considerably lower, with overall Pass@1 accuracy at 0.3756, improving to 0.5134 at Pass@2 and 0.5893 at Pass@3. C++17 consistently outperforms other languages across all k values, achieving the highest scores of 0.3862, 0.5294, and 0.6279 for Pass@1, Pass@2, and Pass@3 respectively. While multiple sampling provides consistent improvements, the absolute performance remains limited compared to Task 1, highlighting the greater difficulty of time complexity assessment.

Table 4: Pass@$k$ Evaluation Results of GPT-4-Turbo. The table shows Pass@$k$ accuracy across different programming languages on Task 1 and Task 3.

| Language | Task 1 | | | Task 3 | | |
| --- | --- | --- | --- | --- | --- | --- |
| | Pass@1 | Pass@2 | Pass@3 | Pass@1 | Pass@2 | Pass@3 |
| Java | **0.6765** | **0.8235** | 0.9118 | 0.3534 | 0.5141 | 0.5904 |
| Python | 0.6000 | 0.7286 | 0.8857 | 0.3793 | 0.5090 | 0.5790 |
| C++17 | 0.6265 | 0.7679 | **0.9442** | **0.3862** | **0.5294** | **0.6279** |
| C++20 | 0.5886 | 0.7275 | 0.9346 | - | - | - |
| C++23 | 0.5862 | 0.7385 | 0.8994 | - | - | - |
| **Overall** | **0.6057** | **0.7493** | **0.9271** | **0.3756** | **0.5134** | **0.5893** |

Table 5: Task 1 Performance Under Different Prompting Strategies for Claude-3-5-Haiku and Qwen2.5-7B-Instruct-1M

| multirow2*Prompt Type | Claude-3-5-Haiku | | Qwen2.5-7B-Instruct-1M | |
|---|---|---|---|---|
| | Accuracy ↑ | Cross-Entropy ↓ | Accuracy ↑ | Cross-Entropy ↓ |
| Default | 52.57 | 0.8202 | 50.00 | 1.0060 |
| Tree-of-Thought | 51.64 | 1.0035 | 51.29 | 1.7278 |
| Chain-of-Draft | 54.36 | 0.6848 | 51.57 | 4.2554 |
| Few-shot | 55.36 | 1.2382 | 51.79 | 4.7582 |
| Chain-of-Thought | **55.50** | **1.5145** | **52.00** | **4.7800** |

## 4.4 PROMPTING STRATEGY IMPACT

Based on previous results, Claude-3-5-Haiku exhibited the lowest accuracy among closed-source models, while Qwen2.5-7B-Instruct-1M showed the poorest performance among open-source models. To investigate potential improvements for Task 1, we systematically evaluated four advanced prompting strategies, including Few-shot, Chain-of-Thought (Wei et al., 2022), Tree-of-Thought (Yao et al., 2023), and Chain-of-Draft (Xu et al., 2025). The detailed prompts are provided in Appendices D.4.2, D.4.3, D.4.4, and D.4.5.

As shown in Table 5, the impact of advanced prompting strategies varied between models and across different approaches. For Qwen2.5-7B-Instruct-1M, all advanced prompting strategies consistently improved accuracy over the default baseline, with Chain-of-Thought achieving the highest improvement to 52.00%. However, Claude-3-5-Haiku exhibited mixed results, with accuracy of Tree-of-Thought decreasing to 51.64%, while the other three strategies showed improvements, with Chain-of-Thought reaching the peak accuracy of 55.50%.

Among the strategies that improved accuracy, Chain-of-Draft emerged as the most balanced approach for Claude-3-5-Haiku, achieving accuracy of 54.36% while maintaining the lowest cross-entropy of 0.6848 among advanced strategies. Although Chain-of-Thought maximizes accuracy across both models, it significantly compromises confidence calibration, producing the highest cross-entropy values despite accuracy gains. For deployment scenarios prioritizing reliable uncertainty quantification, our findings suggest adopting Chain-of-Draft for Claude-3-5-Haiku and Tree-of-Thought for Qwen2.5-7B-Instruct-1M, as these strategies achieve the most favorable accuracy-calibration trade-offs within their respective model families.

## 5 CONCLUSION

We introduce CodeInsightBench, a novel benchmark designed to evaluate deeper code understanding in Large Language Models (LLMs). CodeInsightBench assesses models across three critical tasks including Semantic Code Judgment, Debugging Path Tracking, and Code Efficiency Comparison. Our evaluation of 22 state-of-the-art models (11 open-source and 11 closed-source) reveals a significant performance gap. While the best-performing model GPT-5 achieves strong accuracy (93.43%) in distinguishing correct code versions, performance drops substantially to 45.50% on efficiency analysis tasks requiring deeper algorithmic reasoning. These results provide quantitative evidence that current LLMs excel at surface-level syntactic tasks but struggle with semantic code comprehension and execution flow analysis. The consistent failure patterns across models indicate **fundamental limitations in transitioning from pattern recognition to deep code reasoning.** Our findings establish that advancing LLM code understanding requires moving beyond syntactic accuracy toward robust semantic reasoning and algorithmic analysis capabilities.

**Statement.** All datasets used in this work are publicly available and were obtained through legitimate channels, ensuring no ethical concerns regarding data access or privacy. Large language models were used in this work primarily for grammar correction and language polishing, with details provided in Appendix A. To ensure reproducibility, all code and data are made available as described in Appendix B.

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

APPENDIX CONTENTS

## A    LLM Usage

Large language models (LLMs) were used in this work solely for grammar correction and language polishing to improve the clarity and readability of the manuscript. Specifically, LLMs were employed to (1) correct grammatical errors and improve sentence structure, (2) enhance word choice and phrasing for better clarity, and (3) ensure consistent writing style throughout the paper.

LLMs were not used for research ideation, experimental design, data analysis, result interpretation, or generation of scientific content. All research contributions, including benchmark construction, methodological innovations, experimental results, and scientific conclusions, are entirely the work of the human authors. No LLM-generated content was directly incorporated into the core research findings or claims.

The authors take full responsibility for all content in this paper and have verified the accuracy of all claims and statements. All LLM-assisted text has been thoroughly reviewed and validated by the human authors to ensure factual correctness and appropriateness.

## B    Reproducibility Statement

We have released our data and code on an anonymous website `https://anonymous.4open.science/r/CodeInsightBench-C2C5/`. The repository includes the complete CodeInsightBench dataset, baseline model implementations, evaluation scripts, and experimental results. All materials are provided with detailed documentation to ensure reproducibility and facilitate future research in code comprehension evaluation.

## C    CodeInsightBench Analysis

### C.1    Descriptions of Selected Codeforces Problems

In the context of competitive programming, various problem categories require distinct approaches and strategies. The data presented in Table 6 provides information about 14 selected problems from Codeforces. These problems represent a broad spectrum of algorithmic challenges, encompassing various categories such as Brute Force, Constructive Algorithms, Graph Matching, Implementation, Greedy Algorithms, Mathematics, and Dynamic Programming (DP). Each problem in the table is classified into one or more of these categories based on the type of approach or algorithm required to solve it. This diverse selection highlights the comprehensiveness of the dataset, capturing a wide array of problem types and algorithmic techniques that are central to competitive programming.

Table 6: Details of the 14 Selected Codeforces Problems

| Question ID | Question Description | Problem Tags |
|---|---|---|
| 2049B - pspspsps | Cats are attracted to 'pspspsps', but Evirir, being a dignified dragon, is only attracted to 'pspspsps' with oddly specific requirements. Given a string $s = s_1 s_2 \ldots s_n$ of length $n$ consisting of characters 'p', 's', and '.' (dot), determine whether a permutation* $p$ of length $n$ exists, such that for all integers $i$ ($1 \le i \le n$): 
 - If $s_i$ is 'p', then $[p_1, p_2, \ldots, p_i]$ forms a permutation (of length $i$); 
 -If $s_i$ is 's', then $[p_i, p_{i+1}, \ldots, p_n]$ forms a permutation (of length $n - i + 1$); 
 -If $s_i$ is '.', then there is no additional restriction. | Brute force Constructive algorithms Graph matching Implementation |

**Table 6 – continued from previous page**

| Question ID | Question Description | Problem Tags |
|---|---|---|
| 2049C - MEX Cycle | Evirir the dragon has many friends. They have 3 friends! That is one more than the average dragon.
You are given integers $n$, $x$, and $y$. There are $n$ dragons sitting in a circle. The dragons are numbered $1, 2, \ldots, n$. For each $i$ $(1 \leq i \leq n)$, dragon $i$ is friends with dragon $i-1$ and $i+1$, where dragon 0 is defined to be dragon $n$ and dragon $n+1$ is defined to be dragon 1. Additionally, dragons $x$ and $y$ are friends with each other (if they are already friends, this changes nothing). Note that all friendships are mutual. Output $n$ non-negative integers $a_1, a_2, \ldots, a_n$ such that for each dragon $i$ $(1 \leq i \leq n)$, the following holds:
Let $f_1, f_2, \ldots, f_k$ be the friends of dragon $i$. Then:

$$a_i = \text{mex}(a_{f_1}, a_{f_2}, \ldots, a_{f_k})$$ | Brute force
Constructive algorithms
Greedy
Implementation |
| 2050B - Transfusion | You are given an array $a$ of length $n$. In one operation, you can pick an index $i$ from 2 to $n-1$ inclusive, and do one of the following actions:
1. Decrease $a_{i-1}$ by 1, then increase $a_{i+1}$ by 1. 2. Decrease $a_{i+1}$ by 1, then increase $a_{i-1}$ by 1.
After each operation, all the values must be non-negative. Can you make all the elements equal after any number of operations? | Brute force
Greedy
Math |
| 2050C - Uninteresting Number | You are given a number $n$ with a length of no more than $10^5$.
You can perform the following operation any number of times: choose one of its digits, square it, and replace the original digit with the result. The result must be a digit (that is, if you choose the digit $x$, then the value of $x^2$ must be less than 10).
Is it possible to obtain a number that is divisible by 9 through these operations? | Brute force
DP
Math |

**Table 6 – continued from previous page**

| Question ID | Question Description | Problem Tags |
|---|---|---|
| 2050D - Digital string maximization | You are given a string $s$, consisting of digits from 0 to 9. In one operation, you can pick any digit in this string, except for 0 or the leftmost digit, decrease it by 1, and then swap it with the digit to the left of the picked digit.
For example, in one operation from the string '1023', you can get '1103' or '1022'. Find the lexicographically maximum string you can obtain after any number of operations. | Brute force
Greedy
Math
Strings |
| 2050E - Three Strings | You are given three strings: $a$, $b$, and $c$, consisting of lowercase Latin letters. The string $c$ was obtained in the following way: - At each step, either string $a$ or string $b$ was randomly chosen, and the first character of the chosen string was removed from it and appended to the end of string $c$, until one of the strings ran out. After that, the remaining characters of the non-empty string were added to the end of $c$. - Then, a certain number of characters in string $c$ were randomly changed.
For example, from the strings $a = $ abra and $b = $ cada, without character replacements, the strings caabdraa, abracada, acadabra could be obtained.
Find the minimum number of characters that could have been changed in string $c$. | DP
Implementation
Strings |
| | Continued on next page | |

**Table 6 – continued from previous page**

| Question ID | Question Description | Problem Tags |
|---|---|---|
| 2052A - Adrenaline Rush | Alice's friend is a big fan of the Adrenaline Rush racing competition and always strives to attend every race. However, this time, Alice is the one watching the race. To ensure her friend does not miss any important details, Alice decides to take notes on everything that happens on the track. The first thing Alice notices before the race begins is the numbering of the cars. All the cars line up in front of the starting line in a specific order. The car closest to the line is numbered 1, the second car is numbered 2, and so on, up to the last car, which is numbered $n$. How convenient! — Alice thought. The race begins with the countdown: "Three! Two! One! Go!". Alice observes that the cars start in their original order. However, as the race progresses, their order changes. She records whenever one car overtakes another, essentially swapping places with it on the track. During the race, Alice notices something curious: no car overtakes another more than once. In other words, for any two cars $x$ and $y$, there are at most two overtakes between them during the race: "$x$ overtakes $y$" and/or "$y$ overtakes $x$". At the end of the race, Alice carefully writes down the final order of the cars $c_1, c_2, \ldots, c_n$, where $c_1$ represents the winner of the race. Alice's friend, however, is only interested in the final ranking and discards all of Alice's notes except for the final ordering. As Alice is quite curious, she wonders: What is the longest possible sequence of overtakes she could have observed during the race? Your task is to help Alice answer this question. | Constructive algorithms |

**Table 6 – continued from previous page**

| Question ID | Question Description | Problem Tags |
|---|---|---|
| 2057B - Gorilla and the Exam | Due to a shortage of teachers in the senior class of the "T-generation", it was decided to have a huge male gorilla conduct exams for the students. However, it is not that simple; to prove his competence, he needs to solve the following problem.
For an array $b$, we define the function $f(b)$ as the smallest number of the following operations required to make the array $b$ empty:
1. Take two integers $l$ and $r$, such that $l \le r$, and let $x$ be the $\min(b_l, b_{l+1}, \ldots, b_r)$; 2. Remove all such $b_i$ that $l \le i \le r$ and $b_i = x$ from the array. The deleted elements are removed, and the indices are renumerated.
You are given an array $a$ of length $n$ and an integer $k$. No more than $k$ times, you can choose any index $i$ ($1 \le i \le n$) and any integer $p$, and replace $a_i$ with $p$.
Help the gorilla to determine the smallest value of $f(a)$ that can be achieved after such replacements. | Greedy
Sortings |
| 2057C - Trip to the Olympiad | In the upcoming year, there will be many team olympiads, so the teachers of "T-generation" need to assemble a team of three pupils to participate in them. Any three pupils will show a worthy result in any team olympiad. But winning the olympiad is only half the battle; first, you need to get there...
Each pupil has an independence level, expressed as an integer. In "T-generation", there is exactly one student with each independence level from $l$ to $r$, inclusive. For a team of three pupils with independence levels $a$, $b$, and $c$, the value of their team independence is equal to:

$$(a \oplus b) + (b \oplus c) + (a \oplus c)$$

where $\oplus$ denotes the bitwise XOR operation.
Your task is to choose any trio of students with the maximum possible team independence. | Bitmasks
Constructive algorithms
Greedy
Math |
| 2060D - Subtract Min Sort | You are given a sequence $a$ consisting of $n$ positive integers.
You can perform the following operation any number of times:
- Select an index $i$ ($1 \le i < n$), and subtract $\min(a_i, a_{i+1})$ from both $a_i$ and $a_{i+1}$.
Determine if it is possible to make the sequence non-decreasing by using the operation any number of times. | Greedy |
| | | Continued on next page |

**Table 6 – continued from previous page**

| Question ID | Question Description | Problem Tags |
|---|---|---|
| 2061D - Kevin and Numbers | Kevin wrote an integer sequence $a$ of length $n$ on the blackboard. Kevin can perform the following operation any number of times: - Select two integers $x, y$ on the blackboard such that $|x - y| \leq 1$, erase them, and then write down an integer $x + y$ instead. Kevin wants to know if it is possible to transform these integers into an integer sequence $b$ of length $m$ through some sequence of operations. Two sequences $a$ and $b$ are considered the same if and only if their multisets are identical. In other words, for any number $x$, the number of times it appears in $a$ must be equal to the number of times it appears in $b$. | Bitmasks Data structures |
| 2062C - Cirno and Operations | Cirno has a sequence $a$ of length $n$. She can perform either of the following two operations for any (possibly, zero) times unless the current length of $a$ is 1: 1. **Reverse the sequence.** Formally, $[a_1, a_2, \ldots, a_n]$ becomes $[a_n, a_{n-1}, \ldots, a_1]$ after the operation. 2. **Replace the sequence with its difference sequence.** Formally, $[a_1, a_2, \ldots, a_n]$ becomes $[a_2 - a_1, a_3 - a_2, \ldots, a_n - a_{n-1}]$ after the operation. Find the maximum possible sum of elements of $a$ after all operations. | Brute force Math |

**Table 6 – continued from previous page**

| Question ID | Question Description | Problem Tags |
|---|---|---|
| 2063B - Subsequence Update | After Little John borrowed expansion screws from auntie a few hundred times, eventually she decided to come and take back the unused ones. But as they are a crucial part of home design, Little John decides to hide some in the most unreachable places — under the eco-friendly wood veneers.
You are given an integer sequence $a_1, a_2, \ldots, a_n$, and a segment $[l, r]$ $(1 \le l \le r \le n)$.
You must perform the following operation on the sequence exactly once:
- Choose any **subsequence*** of the sequence $a$, and reverse it. Note that the subsequence does not have to be contiguous. Formally, choose any number of indices $i_1, i_2, \ldots, i_k$ such that $1 \le i_1 < i_2 < \ldots < i_k \le n$. Then, change the $i_x$-th element to the original value of the $i_{k-x+1}$-th element simultaneously for all $1 \le x \le k$. Find the minimum value of $a_l + a_{l+1} + \ldots + a_{r-1} + a_r$ after performing the operation.
* A sequence $b$ is a subsequence of a sequence $a$ if $b$ can be obtained from $a$ by the deletion of several (possibly, zero or all) elements from arbitrary positions. | Constructive algorithms
Data structures
Greedy
Sortings |
| 2063C - Remove Exactly Two | Recently, Little John got a tree from his aunt to decorate his house. But as it seems, just one tree is not enough to decorate the entire house. Little John has an idea. Maybe he can remove a few vertices from the tree. That will turn it into more trees! Right?
You are given a tree of $n$ vertices. You must perform the following operation exactly twice:
1. Select a vertex $v$; 2. Remove all edges incident to $v$, and also the vertex $v$.
Please find the maximum number of connected components after performing the operation exactly twice.
Two vertices $x$ and $y$ are in the same connected component if and only if there exists a path from $x$ to $y$. For clarity, note that the graph with 0 vertices has 0 connected components by definition. | Brute force
Data structures
DFS and similar
DP
Graphs
Greedy
Sortings
Trees |

## C.2 Overview of the CodeInsightBench Dataset

Table 7 presents a comprehensive overview of the CodeInsightBench dataset, which comprises three distinct tasks with varying complexity and scope. The dataset contains 14,035 question-answer pairs in total, distributed across Task 1 (Semantic Code Judgment, 1,400 pairs), Task 2 (Debugging Path Tracking, 3,314 pairs), and Task 3 (Code Efficiency Comparison, 9,321 pairs). Each sample is

constructed from real-world code submissions and designed to support long-context reasoning capabilities, with input lengths spanning from 700 to over 44,000 tokens. This substantial variation in context length, combined with the diverse task formulations, enables comprehensive evaluation of language models beyond mere syntactic understanding, encompassing deep semantic comprehension, logical inference capabilities, and performance-aware code analysis.

Table 7: Statistical Overview of Tasks in CodeInsightBench Dataset

| Task | Task Name | Q&A Pairs | Users | Min Tokens | Max Tokens | Languages |
|------|-----------|-----------|-------|------------|------------|-----------|
| Task 1 | Semantic Code Judgment | 1400 | 1250 | 729 | 29728 | C++17, Python, C++20, Java, C++23, Other |
| Task 2 | Debugging Path Tracking | 3314 | 3093 | 718 | 44089 | C++17, Python, C++20, Java, C++23, other |
| Task 3 | Code Efficiency Comparison | 9321 | 1928 | 739 | 16360 | C++17, Java, Python |

## C.3 Character-Level Edit Distance Analysis in Task 1

Figure 6 visualizes the character-level edit distance distribution, revealing that over 60% of submission pairs differ by fewer than 200 characters. This distribution underscores the task's emphasis on fine-grained semantic reasoning rather than surface-level textual differences. The minimal character variations between correct and incorrect submissions create a challenging evaluation scenario where success depends on deep code comprehension rather than pattern matching or superficial heuristics.

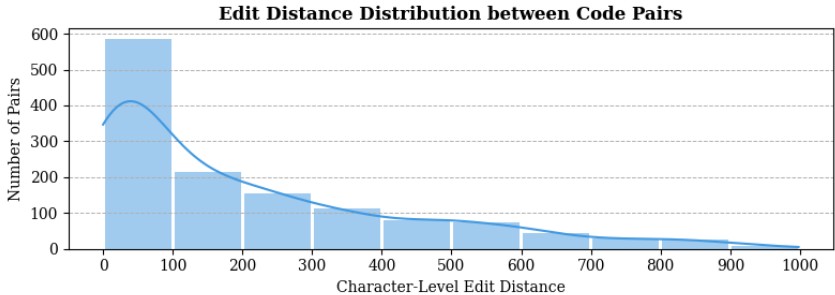

Figure 6: Edit Distance Distribution between Code Pairs

## C.4 Evaluation Metrics

To evaluate the performance of Large Language Models across the three tasks in CodeInsightBench, we employ four main metrics: Accuracy, Cross-Entropy, Pairwise Accuracy, and Pass@$k$.

**Accuracy(Acc.)** measures the proportion of correctly predicted instances among all evaluated examples. If the model prediction matches the ground-truth label, it is considered correct. Higher accuracy indicates better overall predictive performance. It is defined as:

$$\text{Accuracy} = \frac{1}{N} \sum_{i=1}^{N} \mathbb{I}(\hat{y}_i = y_i) \tag{1}$$

where $\hat{y}_i$ is the prediction, $y_i$ is the ground truth, and $\mathbb{I}(\cdot)$ is the indicator function that returns 1 when the prediction is correct.

**Cross-Entropy(CE)** measures the difference between the predicted probability distribution and the true distribution. It is commonly used in classification tasks to evaluate how well the predicted

probabilities align with the actual labels. Lower values indicate better alignment. It is defined as:

$$\text{Cross} - \text{Entropy} = -\frac{1}{N} \sum_{i=1}^{N} [y_i \log(\hat{p}_i) + (1 - y_i) \log(1 - \hat{p}_i)] \tag{2}$$

where $\hat{p}_i$ is the predicted probability for class 1, and $y_i \in \{0, 1\}$ is the ground-truth label.

**Pairwise-Accuracy(Pair Acc.)** evaluates the model's ability to maintain the correct relative ranking between pairs of items. It considers all item pairs in the predicted order and checks whether their order matches the ground-truth ranking. The score is calculated as:

$$\text{Pairwise-Accuracy} = \frac{1}{T} \sum_{i<j} \mathbb{I} \left( \text{GoldRank}[\text{pred}_i] < \text{GoldRank}[\text{pred}_j] \right) \tag{3}$$

where $T = \binom{N}{2}$ is the total number of distinct item pairs, $\text{pred}_i$ is the $i$-th item in the model-predicted ranking, and $\text{GoldRank}[\cdot]$ denotes the position of an item in the ground-truth order.

Pass@$k$ evaluates whether the correct answer appears within the top-$k$ predictions of the model. This metric is particularly useful in code generation or retrieval tasks where multiple outputs can be considered. Higher Pass@$k$ indicates better performance under multiple-choice or top-k generation settings.

$$\text{Pass@}k = \frac{1}{N} \sum_{i=1}^{N} \mathbb{I} \left[ y_i \in \text{TopK}(x_i, k) \right] \tag{4}$$

where $N$ is the total number of samples, $x_i$ is the input for the $i$-th sample, $y_i$ is the ground-truth answer, $\text{TopK}(x_i, k)$ denotes the top-$k$ predictions, and $\mathbb{I}[\cdot]$ is the indicator function that returns 1 if the condition is true, and 0 otherwise.

## D  EXPERIMENT SETUP

### D.1  EVALUATED MODEL CONFIGURATION

Table 8 details the configuration of the 22 models. All model evaluations on CodeInsightBench were conducted using a consistent set of inference parameters. The temperature was uniformly set to 0.7, and the maximum number of output tokens was set to 4096. Inference for closed-source models and DeepSeek models (DeepSeek-V2, DeepSeek-V3) was performed via API calls, while other open-source models were evaluated using local deployments. All inference processes were conducted on NVIDIA A800-SXM4-80GB GPUs. For the detailed Pass@$k$ performance analysis in the Semantic Code Judgment task and the Code Efficiency Comparison task, OpenAI's GPT-4-Turbo was used to generate 3 independent samples per instance to evaluate Pass@$k$ metrics.

### D.2  VISUALIZATION OF MODEL PERFORMANCE IN CLOSED-SOURCE VS. OPEN-SOURCE

Figure 7 presents a comparative bar chart illustrating the accuracy of various Large Language Models (LLMs) on the three distinct tasks within the CodeInsightBench: Semantic Code Judgment, Debugging Path Tracking, and Code Efficiency Comparison. The left panel displays the performance of closed-source models, while the right panel shows the performance of open-source models.

### D.3  VISUALIZATION OF ACCURACY COMPARISON ACROSS LANGUAGES FOR DIFFERENT TASKS

Figure 8 presents a bar chart comparing the accuracy across different programming languages for three distinct tasks. Relative performance among languages varies by task; for instance, Python and C++17 show strong results in Semantic Code Judgment, while Java demonstrates competitive accuracy in Debugging Path Tracking and Code Efficiency Comparison, highlighting task-specific strengths and weaknesses.

Table 8: Model List with Citations and Versions

| Model | Citation | Version |
|-------|----------|---------|
| **Close-Sourced Models** | | |
| Gemini-2.0-Flash-Exp | DeepMind (2025) | Gemini-2.0-Flash-Exp |
| GPT-3.5-Turbo | OpenAI (2025) | gpt-3.5-turbo-1106 |
| GPT-4-Turbo | OpenAI (2025) | gpt-4-turbo |
| GPT-4o | OpenAI (2025) | gpt-4o-2024-08-06 |
| o1-mini | OpenAI (2025) | o1-mini-2024-09-12 |
| o4-mini | OpenAI (2025) | o4-mini-2025-04-16 |
| GPT-4.1-Mini | OpenAI (2025) | gpt-4.1-mini-2025-04-14 |
| GPT-5 | OpenAI (2025) | gpt-5-2025-08-07 |
| Claude-3-5-Haiku | Anthropic (2025) | Claude-3-5-Haiku-20241022 |
| Claude-3-7-Sonnet | Anthropic (2025) | Claude-3-7-Sonnet-20250219 |
| Claude Opus 4.1 | Anthropic (2025) | Claude-Opus-4-1-20250805 |
| GLM-4-Plus | BigModel (2025) | glm-4-plus |
| **Open-Sourced Models** | | |
| Phi-4 | Abdin et al. (2024) | microsoft/phi-4 |
| Llama-3.1-8B-Instruct | Grattafiori et al. (2024) | meta-llama/Llama-3.1-8B-Instruct |
| Gemma-2-9B-It | Team et al. (2024) | google/gemma-2-9b-it |
| Deepseek-V3 | DeepSeek-AI (2024a) | deepseek-ai/DeepSeek-V3 |
| DeepSeek-V2 | DeepSeek-AI (2024b) | deepseek-ai/DeepSeek-V2-Chat |
| Qwen2.5-Coder-7B-Instruct | Hui et al. (2024) | Qwen/Qwen2.5-Coder-7B-Instruct |
| Qwen2.5-7B-Instruct | Yang et al. (2024) | Qwen/Qwen2.5-7B-Instruct |
| Qwen2.5-7B-Instruct-1M | Yang et al. (2025b) | Qwen/Qwen2.5-7B-Instruct-1M |
| Qwen2.5-14B-Instruct-1M | Yang et al. (2025b) | Qwen/Qwen2.5-14B-Instruct-1M |
| Qwen3-8B | Yang et al. (2025a) | Qwen/Qwen3-8B |
| Qwen3-14B | Yang et al. (2025a) | Qwen/Qwen3-14B |

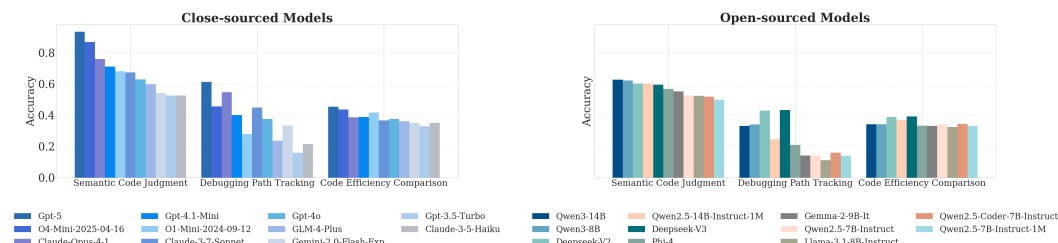

Figure 7: Bar Chart of Model Performance in Closed-Source vs. Open-Source

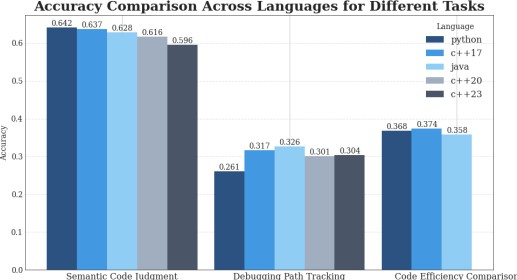

Figure 8: Bar Chart of Accuracy Comparison Across Languages for Different Tasks

## D.4 PROMPT TEMPLATES

Each task in CodeInsightBench has a default prompt template designed to guide the model's response, with varying levels of detail and instructions, along with specific output format requirements. To further analyze the impact of different prompt strategies on model performance, we additionally employed Few-shot, Chain-of-Thought (CoT), Tree-of-Thought (ToT), and Chain-of-Draft (CoD) prompting strategies for evaluation on Task 1. The following subsections provide the detailed prompt information used in the evaluation.

### D.4.1 DEFAULT PROMPTS FOR THREE TASK TYPES

---

**Prompt used for Task 1**

Below is a programming problem, followed by two code submissions. One of them is correct. Please determine the probability that Code1 is correct. The probability should be a value between 0 and 1, rounded to two decimal places. Do not provide any additional explanation or details.

Problem: {question}
Example: {example}

Code1:{code1}
Code2:{code2}

Probability that Code1 is correct:

---

**Prompt used for Task 2**

You are a code review expert tasked with analyzing a series of code submissions to determine their logical order. You are given a problem and out-of-order code submission history. Each submission is marked with a unique ID in the format [ID: xxxx].

Your task is carefully analyze the code under each ID to determine the most likely order they were written in, from the earliest (typically more incorrect) to the latest (typically more correct or refined).Consider factors such as syntax errors, logic issues, completeness, and quality improvements. Provide a JSON array listing the IDs in the inferred logical order. Only output the ID strings in order, like this:["abc123", "xyz789"]

Problem: question
Submissions:{submission_list}

Output:

---

**Prompt used for Task 3**

Compare two correct solutions to the same problem, written in the same language. Focus on their runtime performance based on both the provided code and execution times.

Problem: {question}
Example: {example}

Code1:{code1}
Code2:{code2}

Which code is more efficient?

---

> The answer is: Only return one number: 1 if Code1 is faster, 2 if Code2 is faster, or 0 if the difference is negligible (¡20%). Do not explain or output anything else.

### D.4.2 FEW-SHOT PROMPT FOR TASK1

**Few-shot prompt used for Task 1**

Below is a programming problem, followed by two code submissions. One of them is correct, and the other may have a verdict of 'Wrong answer', 'Compilation error', 'Runtime error', 'Time limit exceeded', or 'Memory limit exceeded'. Please think carefully and predict the probability that Code1 is correct. The probability should be a value between 0 and 1, rounded to two decimal places. A value of 0.5 indicates uncertainty, so try to avoid outputting 0.5. If the probability is less than 0.5, it means Code2 is more likely to be correct. Do not provide any additional explanation or details.
Problem: Check if a number is prime.
Example 1:
Code1:

```
def is_prime(n):
    if n <= 1:
        return False
    for i in range(2, int(n ** 0.5) + 1):
        if n % i == 0:
            return False
    return True
```

Code2:

```
def is_prime(n):
    if n <= 1:
        return True  # Incorrect
    for i in range(2, n):
        if n % i == 0:
            return False
    return True
```

Probability that Code1 is correct: 1.00

Problem: Compute the sum of elements in a list.
Example 2:
Code1:

```
def sum_list(arr):
    total = 0
    for x in arr:
        total += x
    return total
```

Code2:

```
def sum_list(arr):
    total = 0
    i = 0
    while i < len(arr):
        total += arr[i]
        # missing i += 1 causes an infinite loop
    return total
```

Probability that Code1 is correct: 1.00

Problem: {question}
Example: {example}

Code1: {code1}
Code2: {code2}

Probability that Code1 is correct:

### D.4.3 CHAIN-OF-THOUGHT PROMPT FOR TASK1

**Chain-of-Thought prompt used for Task 1**

Below is a programming problem, followed by two code submissions. One of them is correct, and the other may have a verdict of 'Wrong answer', 'Compilation error', 'Runtime error', 'Time limit exceeded', or 'Memory limit exceeded'. Please think carefully and predict the probability that Code1 is correct. The probability should be a value between 0 and 1, rounded to two decimal places. A value of 0.5 indicates uncertainty, so try to avoid outputting 0.5. If the probability is less than 0.5, it means Code2 is more likely to be correct. Do not provide any additional explanation or details. Break down your reasoning into clear steps to make your decision, and evaluate the logic of the code in both submissions.

Problem: {question}
Example: {example}

Code1:{code1}
Code2:{code2}

Thinking step 1: Evaluate the core logic of Code1 and Code2. What does each code do to solve the problem? Is the logic correct?

Thinking step 2: Identify if there are any errors in Code1 and Code2, such as logical or boundary condition errors. Does either code return incorrect results for specific edge cases?

Thinking step 3: Based on the previous steps, decide which code is more likely to be correct. Take into account the logic and correctness of both submissions.

Probability that Code1 is correct:

### D.4.4 TREE-OF-THOUGHT PROMPT FOR TASK1

**Tree-of-Thought prompt used for Task 1**

Below is a programming problem, followed by two code submissions. Imagine three experts analyzing the codes through structured reasoning paths:
- Path 1 (Algorithm Logic): Does the code implement the correct approach for the problem? Vote Code1 or Code2.
- Path 2 (Edge Case Handling): Which code better handles boundary conditions (empty inputs, extreme values, etc.)? Vote accordingly.
- Path 3 (Syntax and Runtime Safety): Does the code have compilation errors, runtime crashes, or infinite loops? Vote the safer one.
- Path 4 (Efficiency): Does the code avoid time/memory limit issues (for applicable problems)? Vote the more efficient one.
Each valid vote for Code1 adds 0.25 to its score; each for Code2 adds 0.25. Total score is normalized to a probability (0.00-1.00), rounded to two decimals. Avoid 0.5 by leaning into reasoning biases. Do not provide any additional explanation or details.

Problem: {question}
Example: {example}

Code1:{code1}
Code2:{code2}

Probability that Code1 is correct:

### D.4.5 CHAIN-OF-DRAFT PROMPT FOR TASK1

> **Chain-of-Draft prompt used for Task 1**
>
> You are given a programming problem and two code submissions. One is correct, the other has a bug. Think in three micro-drafts. Each draft must be less than 5 words. At the end, output the probability that Code1 is correct (rounded to two decimals). Output only the three drafts and the final probability. No explanations.
>
> Problem: {question}
> Example: {example}
>
> Code1:{code1}
> Code2:{code2}
>
> Probability that Code1 is correct:

## E RESULT ANALYSIS

### E.1 IMPACT OF DEBUG PATH LENGTH ON DEBUGGING PATH TRACKING PERFORMANCE

Figure 9 displays model accuracy on the Debugging Path Tracking task, segmented by submission sequence length (Version Count: 3, 4, or 5). A clear trend is the superior performance of several closed-source models, notably o4-mini-2025-04-16 and Claude-3-7-Sonnet, which consistently outperform most open-source counterparts, especially on shorter sequences (Version Count 3, green bars). Large open-source models like Deepseek-V3 and DeepSeek-V2 also demonstrate robust capabilities across all sequence lengths. Generally, accuracy tends to decrease as the submission sequence lengthens from 3 to 5 versions, indicating that tracing longer, more complex debugging paths is more challenging for all models. Many smaller open-source models show markedly lower performance, particularly struggling with longer sequences, highlighting the difficulty of multi-step logical reasoning for these LLMs.

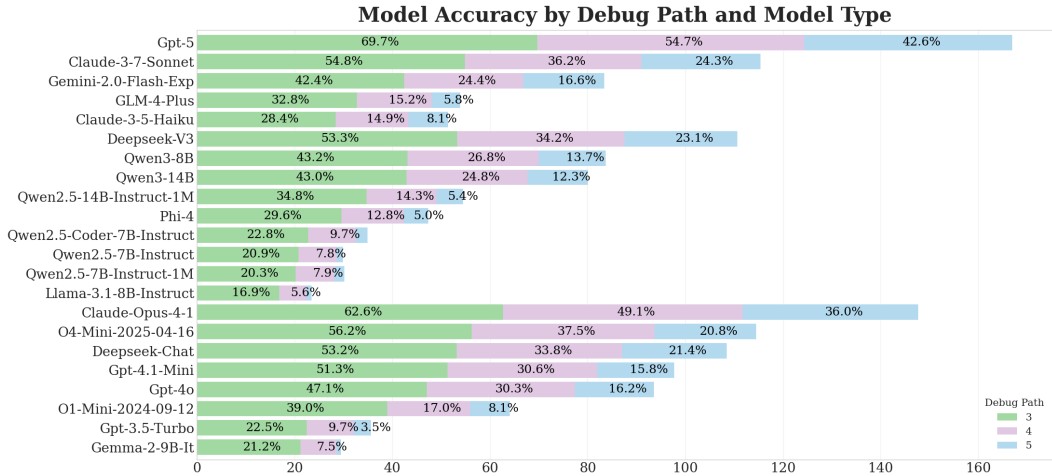

Figure 9: Model Accuracy by Debug Path and Model Type

## E.2 Answer Distribution by Gold Answer Label in Task 3

The Table 9 presents the prediction distribution of various models across different gold answer labels (Gold=0, 1, 2) in Task 3, along with their overall accuracy and standard deviation (STD). It provides a comparative view of both closed-source and open-source models, revealing notable label bias in many models. Specifically, a strong tendency to predict Gold=1 or Gold=2 is observed. The standard deviation reflects the imbalance in predictions across labels, indicating the degree of prediction skewness or bias.

Although the ground-truth labels in the Code Efficiency Comparison task are evenly distributed among the three classes, we observe clear prediction imbalances across many models. For instance, Qwen3-14B predicts label "2" in over 95% of cases, with minimal correct predictions for labels "0" and "1", resulting in a high standard deviation of 0.4231. Similar skewed patterns are observed in models such as Qwen3-8B (std = 0.4297) and Qwen2.5-7B-Instruct-1M (std = 0.3971), suggesting a strong positional bias. In contrast, more capable models like GPT-4o (std = 0.0976) and Qwen2.5-14B (std = 0.1270) exhibit more balanced output distributions. These discrepancies are not caused by the data distribution itself, but rather reflect the models' tendencies to favor specific answer positions—possibly due to format sensitivity or insufficient reasoning ability. This analysis highlights that even when task design is balanced, certain models can still exhibit systematic biases, which in turn limits their effectiveness in nuanced comparative reasoning tasks.

Table 9: Answer Distribution by Gold Answer Label in Task 3

| Type | Model Name | Accuracy | Gold=0 | Gold=1 | Gold=2 | STD |
|------|-----------|----------|--------|--------|--------|-----|
| Closed-source | GPT-5 | 0.4550 | 0.1541 | **0.6283** | 0.6087 | 0.2191 |
| | GPT-4o | 0.3761 | 0.2716 | **0.5075** | 0.3563 | 0.0976 |
| | Claude-Opus-4-1 | 0.3865 | 0.0271 | **0.5101** | 0.6561 | 0.2688 |
| | Claude-3-5-Haiku | 0.3508 | 0.1347 | **0.5093** | 0.4291 | 0.1611 |
| | o1-mini-2024-09-12 | 0.4181 | 0.1256 | 0.5387 | **0.6475** | 0.2248 |
| | GPT-3.5-Turbo | 0.3304 | 0.0341 | 0.4053 | **0.5805** | 0.2278 |
| | o4-mini-2025-04-16 | 0.4373 | 0.1094 | **0.6624** | 0.5726 | 0.2423 |
| | GPT-4.1-mini | 0.3886 | 0.0442 | 0.5363 | **0.6263** | 0.2559 |
| | Claude-3-7-Sonnet | 0.3678 | 0.0037 | 0.4814 | **0.6528** | 0.2747 |
| | Gemini-2.0-Flash-Exp | 0.3509 | 0.0198 | 0.3664 | **0.7005** | 0.2779 |
| | GLM-4-Plus | 0.3619 | 0.0042 | 0.4447 | **0.6849** | 0.2819 |
| Open-source | Qwen2.5-14B-Instruct-1M | 0.3691 | 0.3042 | **0.5439** | 0.2522 | 0.1270 |
| | Qwen2.5-Coder-7B-Instruct | 0.3424 | **0.4793** | 0.4243 | 0.1048 | 0.1651 |
| | Qwen2.5-7B-Instruct | 0.3470 | 0.1832 | 0.2371 | **0.6440** | 0.2057 |
| | Phi-4 | 0.3344 | 0.1185 | **0.6270** | 0.2716 | 0.2130 |
| | Gemma-2-9B-It | 0.3327 | 0.0149 | **0.6897** | 0.3160 | 0.2760 |
| | DeepSeek-V2 | 0.3891 | 0.0385 | **0.7457** | 0.4090 | 0.2888 |
| | DeepSeek-V3 | 0.3920 | 0.0388 | **0.7576** | 0.4052 | 0.2935 |
| | Llama-3.1-8B-Instruct | 0.3251 | 0.0000 | **0.7897** | 0.2057 | 0.3345 |
| | Qwen2.5-7B-Instruct-1M | 0.3425 | 0.0014 | **0.8966** | 0.1195 | 0.3971 |
| | Qwen3-14B | 0.3429 | 0.0030 | 0.1159 | **0.9516** | 0.4231 |
| | Qwen3-8B | 0.3387 | 0.0039 | 0.1009 | **0.9600** | 0.4297 |

## E.3 Code Similarity Calculation Methodology

To quantify the degree of code modification between different submissions in the debugging path tracking analysis, we employ standardized similarity metrics using Python's `difflib` library.

Code similarity is computed using `difflib.SequenceMatcher`, which implements the Ratcliff-Obershelp algorithm. The similarity score is calculated as:

$$\text{Similarity} = \frac{2 \times \text{Number of Matching Characters}}{\text{Length of Code}_1 + \text{Length of Code}_2}$$

This metric ranges from 0 to 1, where 1 indicates identical code and 0 represents completely different code. The algorithm identifies the longest common subsequences between two code snippets and

computes the ratio based on the total matching characters relative to the combined length of both code segments.

To measure the extent of code modifications, we calculate the number of altered lines using `difflib.unified_diff`. This function generates a unified diff format that identifies added, deleted, and modified lines between two code versions. The difference line count represents the total number of lines that differ between the original and modified code submissions.

These metrics provide complementary perspectives on code evolution that similarity scores capture overall structural preservation, while difference line counts quantify the granular extent of modifications. Together, they enable systematic analysis of the relationship between code transformation complexity and model performance degradation.

