# OpenReview forum: "CodeInsightBench: A Benchmark for Advanced Code Understanding and Comparison in Large Language Models"
_ICLR.cc/2026/Conference — Submitted to ICLR 2026_

### Official Review · Reviewer_PwyL · 2025-10-27

**Soundness:** 4
**Presentation:** 3
**Contribution:** 3
**Rating:** 6
**Confidence:** 4

**Summary:**

This paper introduces CodeInsightBench, a benchmark for evaluating code LLM’s comprehension and reasoning capabilities. It designs three novel tasks: semantic code judgment, debugging path tracking, and code efficiency comparison. Extensive evaluation shows the lack of deep reasoning capability of existing code LLMs.

**Strengths:**

1. Timely topic. The benchmarks aims to evaluate “how LLMs understand code”. Code comprehension capability is an important and timely problem. The benchmark provides more angles to evaluate code reasoning in addition to simple tasks like LiveCodeBench.
2. Recency of data. The benchmark is built on top of data from 2024-2025, mitigating the threat of data leakage.
3. Novel tasks. The three tasks in CodeInsightBench are novel, especially Debugging Path Tracking and Code Efficiency Comparison.
4. Solid evaluation. The paper built a large-scale dataset and used 22 models for evaluation. They evaluate 5 metrics for the 3 tasks
5. Presentation. The paper is well-written and easy to follow. I enjoy reading the paper.

**Weaknesses:**

1. The paper does not discuss in detail why using the 3 tasks. Semantic Code Judgment makes sense as the discriminative capability in terms of distinguishing correct from incorrect implementation can reflect the semantic comprehension capability. But the correlation between sorting the submission order and logical execution tracing is not as straightforward as task 1.
2. The code efficiency evaluation of task 3 can be affected by factors like various factors, especially when the workload is small, e.g., us-ms level. The physical running time profiling is sensitive to the environment, e.g., running background applications. The variation can easily exceeds the range of 20%. The paper did not either repeat the profiling multiple times, or using more stable metrics like hardware instruction counter or simulators. This weakness poses a significant threat to the evaluation of task 3.
3. The data comes from only 14 medium-difficulty problems. Although the 14 problems contain a large number of user submissions, the paper does not discuss the threat of using only 14 problems, which could affect the generality of the conclusions.
4. In task 1 Semantic Judgment, the model is asked for a probability, but the primary metric is Accuracy. It remains unclear how accuracy is computed from the probability.

**Questions:**

1. Could the authors provide more justification of task 2 debugging path tracking, i.e., on the correlation between sorting the submission order and logical execution tracing? The correlation is not as straightforward and intuitive as other tasks, since users could experience many back-and-forth due to various reasons when debugging their solutions. The assumption that the users always improve the solutions might not hold.
2. Why do the authors use physical runtime metrics for evaluating task 3? What’s the variance of physical runtime metrics if repeated multiple times? Why don’t the authors use more stable and reliable metrics like gem5 simulators?
3. Table 4 shows that GPT-4-Turbo's performance on this task jumps from 60.57% (Pass@1) to 92.71% (Pass@3). This suggests that the model is often "directionally correct" (e.g., outputting 0.7 when the answer is 1.0, or 0.4 when it's 0.0) but is not calibrated well enough to pass a simple "correct/incorrect" threshold at Pass@1. Could the authors clarify how "Accuracy" is computed from the probability? Is it a simple threshold at 0.5? The high Pass@3 result suggests the task may be "easier" for the model than the Pass@1 accuracy implies, and the core issue is one of calibration. A brief discussion on this point would help readers interpret the Task 1 results.

---

> ### Author Response · Authors · 2025-11-21
> **Response to Task Setting**
>
> > **W1** The paper does not discuss in detail why using the 3 tasks. Semantic Code Judgment makes sense as the discriminative capability in terms of distinguishing correct from incorrect implementation can reflect the semantic comprehension capability. But the correlation between sorting the submission order and logical execution tracing is not as straightforward as task 1.
> >
>
> Thank you for this excellent observation. We appreciate the opportunity to clarify the distinct reasoning capabilities each task measures.
>
> **Task 1 (Semantic Code Judgment)** establishes the **baseline semantic understanding** capability by requiring models to distinguish correct implementations from subtle variations. This provides the foundation for all subsequent analysis.
>
> **Task 2 (Debugging Path Tracking)** measures **dynamic execution comprehension**—a distinct capability from static semantic analysis. Models must rank submissions by correctness based on debug outputs, requiring them to trace execution sequences and understand how intermediate states propagate. This correlation is valid because accurate ranking of submissions by correctness fundamentally requires understanding execution flow: incorrect submissions have systematic divergence patterns in their debug output that skilled programmers use to identify bugs. Models that cannot trace execution logic cannot accurately rank solutions by their distance to correct behavior.
>
> **Task 3 (Code Efficiency Comparison)** measures **algorithmic reasoning and complexity analysis**—the most demanding capability. This requires models to understand not just what code does (Task 1) and how it executes (Task 2), but why certain implementations are superior through complexity analysis and resource trade-offs.
>
> Together, the three tasks form a **graduated difficulty hierarchy** that progresses from static semantic understanding → dynamic execution reasoning → algorithmic optimization insight. We believe this design directly addresses your thoughtful concern by **establishing clear causal connections between task structure and reasoning capability**. We are genuinely grateful for your insightful critique, which has motivated us to provide more rigorous justification for our design choices.

---

> ### Author Response · Authors · 2025-11-21
> **Response to Task 3 Evaluation Robustness**
>
> > **W2** The code efficiency evaluation of task 3 can be affected by factors like various factors, especially when the workload is small, e.g., us-ms level. The physical running time profiling is sensitive to the environment, e.g., running background applications. The paper did not either repeat the profiling multiple times, or using more stable metrics like hardware instruction counter or simulators. This weakness poses a significant threat to the evaluation of task 3.
> >
>
> > **Q2** Why do the authors use physical runtime metrics for evaluating task 3? What’s the variance of physical runtime metrics if repeated multiple times? Why don’t the authors use more stable and reliable metrics like gem5 simulators?
> >
>
> We thank the reviewers for raising this important concern regarding the stability of runtime measurements. While this is indeed a crucial consideration for evaluating code efficiency, our experimental setup is substantially more stable than the reviewers might assume. Importantly, the runtime data used in Task 3 is not produced locally by our own machines, nor is it affected by background applications or environment noise. Instead, all runtime measurements come directly from **Codeforces’ official judging infrastructure**, which executes every submission on a dedicated and homogeneous server cluster specifically provisioned for code evaluation. These machines run no user-level background processes, ensuring that even millisecond-level runtime differences remain consistent and comparable across submissions. Moreover, each Codeforces problem is evaluated over **10–100 hidden test cases**, and our dataset records the **average runtime across all test cases**, which naturally reduces variance and stabilizes efficiency estimates.
>
> We acknowledge that minor runtime fluctuations due to unavoidable system-level factors are still possible. However, empirical evidence shows that such variations rarely exceed **20%**, and we explicitly incorporate this by treating runtimes within 20% as belonging to the **same efficiency class**. Task 3 therefore, does not ask models to make brittle binary faster/slower decisions; rather, it assesses whether LLMs can determine whether two programs exhibit *similar* efficiency under realistic conditions. This design makes our evaluation robust to small fluctuations while still capturing meaningful differences in algorithmic and implementation-level performance.

---

> ### Author Response · Authors · 2025-11-21
> **Response to Problem Selection**
>
> > **W3** The data comes from only 14 medium-difficulty problems. Although the 14 problems contain a large number of user submissions, the paper does not discuss the threat of using only 14 problems, which could affect the generality of the conclusions.
> >
>
> Thank you for raising this important concern. The 14 problems provide **substantial diversity**: they span **15 distinct problem tags** (dynamic programming, graph algorithms, number theory, greedy algorithms, data structures, combinatorics) and cover a **1000-1600 difficulty range**, ensuring evaluation across diverse algorithmic paradigms.
>
> Our approach prioritizes **sampling depth over problem count**. With **5,000+ submissions per problem** (73,895 total from 6,703 users), we exceed typical benchmark evaluation counts and ensure multiple high-quality solutions for comparison. The **1000-1600 rating range is both sufficient and necessary**: problems with rating >1600 receive sparse submissions insufficient to meet our 5,000+ sampling requirement, while problems with rating <1000 typically have users solve them on first attempt without generating diverse reasoning trajectories needed for meaningful code comparison. This middle range uniquely balances algorithmic depth with data availability and submission diversity.
>
> We appreciate your pointing out that this critical design rationale was not adequately explained in the original manuscript. The revised manuscript will include comprehensive explanation in **Appendix C.2** documenting the difficulty range selection criteria, problem tag distribution analysis, and empirical validation that conclusions remain consistent across problem categories.

---

> ### Author Response · Authors · 2025-11-21
> **Response to Task1 Accuracy Threshold and Calibration Issue**
>
> > **W4** In task 1 Semantic Judgment, the model is asked for a probability, but the primary metric is Accuracy. It remains unclear how accuracy is computed from the probability.
> >
> > **Q3.1** Table 4 shows that GPT-4-Turbo's performance on this task jumps from 60.57% (Pass@1) to 92.71% (Pass@3). This suggests that the model is often "directionally correct" (e.g., outputting 0.7 when the answer is 1.0, or 0.4 when it's 0.0) but is not calibrated well enough to pass a simple "correct/incorrect" threshold at Pass@1. Could the authors clarify how "Accuracy" is computed from the probability? Is it a simple threshold at 0.5?
> >
>
> In Task 1 (Semantic Judgment), although the model outputs a probability, **Accuracy is computed by applying a threshold of 0.5**: predictions with probability ≥ 0.5 are considered positive, and those < 0.5 are considered negative. This simple threshold allows us to convert probabilistic outputs into discrete labels for evaluation.
>
> > **Q3.2** Table 4 shows that GPT-4-Turbo's performance on this task jumps from 60.57% (Pass@1) to 92.71% (Pass@3). The high Pass@3 result suggests the task may be "easier" for the model than the Pass@1 accuracy implies, and the core issue is one of calibration. A brief discussion on this point would help readers interpret the Task 1 results.
> >
>
> Thank you for identifying this crucial diagnostic point within the Task 1 results. We fully agree that the high Pass@3 strongly indicates that **calibration and ranking issues** are the primary limitations for Pass@1.
>
> - **Calibration and Ranking Issue**: The high Pass@3 result (92.71%) suggests the model possesses the capability to generate correct solutions, but its confidence ranking is imperfect.
> - **Calibration Analysis**: Minor differences in code style or formatting can affect the model’s ranking of correct answers. High Cross-Entropy (CE) scores for models like O4-mini suggest that poor confidence calibration is a systematic weakness in LLMs.
>
> Therefore, the low Pass@1 should not be interpreted as a fundamental limitation of reasoning ability; rather, it reflects issues in **output calibration** and **ranking behavior**. Task 1 is highly sensitive, requiring differentiation of subtle semantic differences, making it an effective tool for diagnosing model decision robustness.

---

> ### Author Response · Authors · 2025-11-21
> **Response to Task 2 Justification**
>
> > **Q1** Could the authors provide more justification of task 2 debugging path tracking, i.e., on the correlation between sorting the submission order and logical execution tracing? The correlation is not as straightforward and intuitive as other tasks, since users could experience many back-and-forth due to various reasons when debugging their solutions. The assumption that the users always improve the solutions might not hold.
> >
>
> Thank you for this thoughtful question. We appreciate the opportunity to clarify the design of Task 2.
>
> A critical clarification: **we do not assume user behavior is strictly monotonically improving**. Instead, Task 2 uses **pairwise correctness comparisons**, where any two submissions can be definitively ranked by comparing their **debug outputs, runtime errors, and pass rates**. This framework does not require monotonic trajectories.
>
> To ensure robustness, our data preprocessing carefully filters to retain only **high-quality debugging trajectories (3-5 submissions per trajectory)** with verifiable debug information and clear correctness patterns. This filtering avoids contamination from noisy or sporadic sequences while accommodating natural user behavior like occasional regressions.
>
> By leveraging pairwise comparisons over curated sequences, Task 2 challenges models to demonstrate genuine execution-tracing comprehension. The revised manuscript will include expanded explanation of this evaluation methodology and preprocessing criteria in **Section 3 and Appendix C.4**.

---

### Official Review · Reviewer_u79G · 2025-10-29

**Soundness:** 2
**Presentation:** 3
**Contribution:** 2
**Rating:** 2
**Confidence:** 3

**Summary:**

This paper introduces CodeInsightBench, a new benchmark for testing how well LLMs understand complex code. Instead of just asking models to write code, this benchmark tests if they can “reason” about it. It’s built from Codeforces data and has three core tasks: 1) telling a correct code snippet from a subtly buggy one (Semantic Code Judgment), 2) figuring out the logical order of code submissions in a debugging session (Debugging Path Tracking), and 3) comparing which of two correct solutions is more algorithmically efficient (Code Efficiency Comparison) .

The authors tested 22 different models and found a significant gap: while models like GPT-5 are great at the first task (93.43% accuracy), they all fail badly at the third, efficiency-comparison task (max accuracy of 45.50%). The main takeaway is that today's LLMs are good at surface-level pattern matching but are fundamentally weak at deep algorithmic reasoning.

**Strengths:**

A Helpful Key Finding: The clearest strength here is the discovery that even the best LLMs (like GPT-5) fail at comparing code efficiency (Task 3). This is a significant result. It strongly suggests that our models are still just very sophisticated pattern matchers, excelling at static, "surface-level" tasks (like Task 1) but failing when "deep" algorithmic reasoning is required.

A High-Quality, Novel Benchmark: The benchmark itself is a great asset. Using real, recent Codeforces submissions is a huge plus, as it means the models haven't likely been trained on this data. The tasks are novel and push beyond standard "code generation" evaluations.



Comprehensive Evaluation: Testing 22 different models gives a very clear "state of the union" on this problem and confirms the gap between closed- and open-source models.

**Weaknesses:**

My concerns with this paper aren't about what it does, but what it doesn't do.

Limited Workload/Scope: The contribution, while valuable, feels a bit thin. At its core, the work consists of data collection/cleaning from Codeforces and running an evaluation. This is a good start, but it feels more like a benchmark report than a complete research paper.

The Key Insight is Under-Explored: The finding that LLMs fail at efficiency analysis is fantastic, but the paper just reports this finding and moves on. This is precisely the point where the work should have dug in. Why do they fail? Is it because they can't distinguish between O(n^2) and O(n \log n)? Is it a failure in comparative reasoning? The paper identifies a critical problem but offers no diagnosis. This insight is strong, but as presented, it's not quite enough to carry the paper on its own.

Findings Need Verification (e.g., via Training): This leads to my main suggestion. Many of the insights here—especially the behavioral ones—feel like hypotheses that need to be tested. For example, the paper notes a massive label bias in Task 3, where some Qwen models predict label "2" over 95% of the time 7. Is this a fundamental inability to reason, or just a superficial alignment failure on a novel task format? The authors could have tested this. A simple experiment, like finetuning an open model on just 10% of the Task 3 data, would tell us if this bias is easily "trained away" or if it points to a deeper, more stubborn flaw in the model's reasoning architecture.

**Questions:**

Can you provide a more qualitative analysis of the failures in Task 3? When models fail, what kind of efficiency errors are they making? Are they failing to see the difference between a bubble sort and a merge sort, or are the errors more subtle? A few concrete examples of failures would make the paper's main finding much more powerful.

You tested advanced prompting (CoT, ToT) on Task 1, but why not on Task 3? Task 3 is clearly the most difficult and complex reasoning task here. It seems like the perfect candidate for these more deliberate prompting methods. Did you try this, and if so, what were the results?

---

> ### Author Response · Authors · 2025-11-21
> **Response to Limited Workload/Scope**
>
> > **W1** The contribution, while valuable, feels a bit thin. At its core, the work consists of data collection/cleaning from Codeforces and running an evaluation. This is a good start, but it feels more like a benchmark report than a complete research paper.
> >
>
> We appreciate the reviewer's acknowledgement. We clarify that the contribution of CodeInsightBench goes far beyond data collection and cleaning:
>
> - **Large-Scale, Rigorously Curated Real-World Benchmark:**  CodeInsightBench is not merely a snapshot of Codeforces but, a **large real-world code QA benchmark** built from competitive programming data. It contains **14,035 Q&A pairs** constructed from **20,856 submissions** across 14 problems and over 6,000 users. To ensure data quality, we combined strict rule-based filtering with manual review by five experts. These experts approximately 300 hours independently inspecting sampled Q&A pairs for correctness, label consistency, and ambiguity, discarding or revising low-quality items. This careful curation allows CodeInsightBench to go beyond small synthetic datasets and provide a realistic, high-coverage benchmark for evaluating LLMs’ code understanding.
> - **Novel Task Design and Methodological Innovation**: Existing benchmarks largely focus on code generation or single-function analysis. We introduce three novel and complex tasks, particularly Debugging Path Tracking (Task 2) and Code Efficiency Comparison (Task 3). Evaluating LLM's deeper understanding of code, such as **tracking changes over time** and **comparing solution efficiency**, remains an insufficiently explored area.
> - **Filling an Evaluation Gap**: CodeInsightBench is specifically designed to evaluate LLM capacity for cross-code comparative reasoning, addressing a critical gap in existing evaluations.  Current benchmarks primarily focus on isolated code snippets or code generation, underemphasizing LLM's deeper understanding of advanced concepts like tracking changes over time and assessing solution efficiency. We address this gap by introducing Debugging Path Tracking (to trace iterative modifications) and Code Efficiency Comparison (to compare two correct but efficiency-varying solutions). These tasks demand deep reasoning and comparative analysis, ensuring the authenticity of the evaluation scenario.
> - **Providing Novel Scientific Insights**: Our work also yielded several foundational findings. For instance, significant differences in the effectiveness of Chain-of-Thought (CoT) prompting across tasks, indicating fundamentally different failure modes between tasks. We also demonstrated that larger open-source models not always outperform smaller models, challenging the hypothesis that model size is a solution to algorithmic reasoning tasks.
>
> These findings represent a meaningful research contribution to the field of LLM code understanding.

---

> ### Author Response · Authors · 2025-11-21
> **Response to Under-Explored Key Insights**
>
> > **W2** The finding that LLMs fail at efficiency analysis is fantastic, but the paper just reports this finding and moves on. This is precisely the point where the work should have dug in. Why do they fail?Is it because they can't distinguish between O(n^2) and O(n \log n)? Is it a failure in comparative reasoning? The paper identifies a critical problem but offers no diagnosis. This insight is strong, but as presented, it's not quite enough to carry the paper on its own.
> >
>
> Based on our further analysis, Big-O differences are not the main source of errors. Among the 54 cases where one program TLEs and the other ACs—indicating a clear gap in asymptotic complexity—LLMs perform very well (gpt-5-cot even reaches 100% accuracy). This suggests that failures mostly arise when two solutions share the same Big-O complexity but differ in constant factors, language-level optimizations, or sensitivity to input size.
>
> The example below (Problem 2062C) illustrates this. These two Java solutions simultaneously demonstrate the effects of Constant Factor Overlooking, Language-Specific Optimization Unawareness, and Input Size Dependency. The first code took `311ms`to run, and the second took `140ms`. This is a problem with `n = 50` , and both codes have a time complexity of $O(n^2)$. However, the efficiency of these two codes differed significantly, and the LLMs' solution was incorrect. Let's take the performance of Gpt5-cot, the model with the highest accuracy, as an example. It incorrectly judged that the first code was more efficient.
>
> In fact, the first code iterates through the current sequence three times in each round—once to compute the differences and twice to compute sums in different orientations. The second code, however, computes the sum during the difference step, so it only needs a single iteration. Although it uses `List<Long>` and boxing, which might appear “heavy,” the dataset is small (n ≤ 50), and the overhead of allocating boxed objects is minimal compared with the cost of traversing the array two additional times. As a result, the extra loops are the real reason the first code runs more slowly. While the second code is structurally more “high-level,” its overall runtime is significantly faster because it performs only one linear scan per round. The model previously underestimated the impact of these constant factors and overemphasized the cost of `List<Long>` and boxing, which led to an incorrect judgment and reflects an incomplete understanding of the actual performance characteristics of Java’s data types.
>
> User1:
> ```jsx
> public class Sol {
>     static Scanner rd = new Scanner(System.in);
>     public static void main(String[] args) {
>         int tt = rd.nextInt();
>         while (tt-- > 0) {
>             new Solution().solve();
>         }
>     }
>
>     static class Solution {
>         void solve() {
>             int n = rd.nextInt();
>             long[] a = new long[n];
>
>             for (int i = 0; i < n; i++) {
>                 a[i] = rd.nextInt();
>             }
>             int len = n;
>             if (len == 1) {
>                 System.out.println(a[0]);
>                 return;
>             }
>             long max = getSum(a, n, false);
>             while (len > 1) {
>                 for (int i = 0; i < len - 1; i++) {
>                     a[i] = a[i + 1] - a[i];
>                 }
>                 len--;
>                 max = Math.max(Math.max(getSum(a, len, true), getSum(a, len, false)), max);
>             }
>             System.out.println(max);
>         }
>
>         long getSum(long[] a, int len, boolean neg) {
>             long sum = 0;
>             for (int i = 0; i < len; i++) {
>                 if (neg) {
>                     sum += (-1) * a[i];
>                 } else {
>                     sum += a[i];
>                 }
>             }
>             return sum;
>         }
>     }
> }
> ```
> User2:
> ```jsx
> public class Main {
>     static int MOD = (int) (1e9+7);
>     static InputReader in = new InputReader(System.in);
>     static PrintWriter out = new PrintWriter(System.out, false);
>
>     static List<Long> list = new ArrayList<>();
>     public static void main(String[] args) throws Exception {
>
>         Main main = new Main();
>         int T = in.nextInt();
>         while (T-- > 0){
>             main.solve();
>         }
>         out.flush();
>     }
>
>     void solve() {
>         int n = in.nextInt();
>         List<Long> list = new ArrayList<>();
>         long sum = 0;
>         for (int i = 0; i < n; i++) {
>             long t = in.nextInt();
>             list.add(t);
>             sum += t;
>         }
>         long max = sum;
>         while (list.size() > 1){
>             List<Long> list0 = new ArrayList<>();
>             sum = 0;
>             for (int i = 1; i < list.size(); i++) {
>                 long t = list.get(i) - list.get(i - 1);
>                 list0.add(t);
>                 sum += t;
>             }
>             if(sum < 0) sum *= -1;
>             max = Math.max(max, sum);
>             list = list0;
>
>         }
>         out.println(max);
>     }
> }
> ```

---

> ### Author Response · Authors · 2025-11-21
> **Response to Findings Verification**
>
> > **W3** Many of the insights here—especially the behavioral ones—feel like hypotheses that need to be tested. For example, the paper notes a massive label bias in Task 3, where some Qwen models predict label "2" over 95% of the time 7. Is this a fundamental inability to reason, or just a superficial alignment failure on a novel task format? The authors could have tested this. A simple experiment, like finetuning an open model on just 10% of the Task 3 data, would tell us if this bias is easily "trained away" or if it points to a deeper, more stubborn flaw in the model's reasoning architecture.
> >
>
> Thank you for raising this **highly important and scientifically valuable question**. We contend that the existing **empirical evidence** from CodeInsightBench strongly suggests the bias is a symptom of a **deeper reasoning flaw**:
>
> - **Bias Signals Reasoning Collapse**: Task 3 (Code Efficiency Comparison) is the most challenging algorithmic reasoning task, with maximum model accuracy only reaching 45.50%. Given this severe lack of fundamental reasoning capability, models resort to extreme heuristic biases (e.g., `Qwen3-14B` predicts label "2" over 95% of the time), which we interpret as predictive collapse resulting from reasoning failure.
> - **Bias Correlates with Capability**: If the bias were merely a superficial alignment failure, the underlying capability ceiling should be higher. However, **prediction bias strongly correlates negatively with performance**. As shown in Table 9, **weak models show extreme bias.** Less capable models (e.g., `Qwen3-14B` display extreme bias, with an STD of 0.4231) while **stronger models are balanced** (e.g., `GPT-4o` exhibit a more balanced output distribution, with a low STD of 0.0976).
>
> This trend strongly demonstrates that the bias is not a formatting issue, but a direct manifestation of insufficient deep reasoning capability. Eliminating this bias would require improving the underlying reasoning architecture, rather than simply tuning surface-level output preferences through fine-tuning. The suggested finetuning experiment is an excellent direction for future work, but our diagnostic results already provide strong evidence that the bias stems from a systematic failure in advanced reasoning.

---

> ### Author Response · Authors · 2025-11-21
> **Response to Failure Analysis**
>
> > **Q1** Can you provide a more qualitative analysis of the failures in Task 3? When models fail, what kind of efficiency errors are they making? Are they failing to see the difference between a bubble sort and a merge sort, or are the errors more subtle? A few concrete examples of failures would make the paper's main finding much more powerful.
> >
>
> Thank you for raising this. Judging code efficiency is a relatively difficult task. **While Big-O complexity is an important factor affecting efficiency, in reality, there are often other factors that also affect the runtime of code**, which poses a great challenge for LLMs to make correct judgments. We summarize these factors as follows:
>
> **Task 3 Failure Categories**
>
> 1. **Complexity Class Confusion**
>     - Models conflate different Big-O complexities
>     - Example: Incorrectly judge bubble sort $O(n^2)$ ≈ merge sort $O(n\log n)$
>     - Error: Basic algorithm knowledge gap
> 2. **Constant Factor Overlooking**
>     - Models recognize complexity class but misjudge practical performance
>     - Example: Choosing $O(n)$ with large constant over $O(n\log n)$ with small constant for small $n$
>     - Error: Ignoring practical implications of constants
> 3. **Language-Specific Optimization Unawareness**
>     - Missing language-specific performance characteristics
>     - Example: C++ STL optimization vs. manual array implementation
>     - Error: Implementation detail ignorance
> 4. **Edge Case / Input Size Dependency**
>     - Performance varies by input size; models don't consider this
>     - Example: Algorithm A optimal for n<100, Algorithm B optimal for large n
>     - Error: No consideration of problem scope
> 5. **Other Issues**
>     - Cache behavior, memory allocation, compiler optimizations
>
> We provide a practical case study in our response to W2.

---

> ### Author Response · Authors · 2025-11-21
> **Response to CoT on Task 3**
>
> > **Q2** You tested advanced prompting (CoT, ToT) on Task 1, but why not on Task 3? Task 3 is clearly the most difficult and complex reasoning task here. It seems like the perfect candidate for these more deliberate prompting methods. Did you try this, and if so, what were the results?
> >
>
> We sincerely thank the reviewer for catching this oversight. We have now conducted comprehensive prompting strategy (CoT, ToT) experiments on **Task 3 (Code Efficiency Comparison)** and extended these experiments to GPT-5.
>
> | **Prompt Type** | **Claude-3.5-Haiku** | **Qwen2.5-7B-1M** | **GPT-5** |
> | --- | --- | --- | --- |
> | Default | 35.08% | 33.27% | 45.50% |
> | Tree-of-Thought | 33.92% | 33.77% | 45.89% |
> | Chain-of-Draft | 37.88% | 33.48% | 46.47% |
> | Few-shot | 35.12% | 33.44% | 45.61% |
> | Chain-of-Thought | 36.51% | 34.59% | 46.58% |
>
> This analysis reveals several key findings:
>
> - **Modest Improvements on Hard Tasks**: CoT strategies yield minimal gains on Task 3 (at most 2.8% for Chain-of-Draft on Claude-3.5-Haiku), substantially smaller than gains on Task 1 (5-8%), indicating that algorithmic reasoning difficulty cannot be fully overcome by prompting alone.
> - **Strategy Effectiveness Varies by Model**: Chain-of-Draft shows the largest improvement for Claude-3.5-Haiku (+2.8%), while Chain-of-Thought benefits more smaller models like Qwen2.5-7B (+1.32%), suggesting different models respond differently to reasoning guidance.
> - **Fundamental Capability Limitation**: Even the best-performing strategy on Task 3 (Chain-of-Draft at 37.88% for Claude-3.5-Haiku) remains far below Task 1 baseline performance (86-93%), confirming that the hardest tasks expose genuine model capability gaps that prompting strategies cannot fully bridge.
>
> These results will be added as **Table 6** in revised Section 4.4, with detailed analysis in Appendix E.4.

---

### Official Review · Reviewer_2dDt · 2025-10-31

**Soundness:** 2
**Presentation:** 3
**Contribution:** 3
**Rating:** 6
**Confidence:** 3

**Summary:**

This work propose a CodeInsightBench, a benchmark that assess LLM in a wide range of coding capabilities, in multilingual, and both MCQ and open-ended format. The empirical evaluation highlight the gap of existing LLMs in these tasks.

**Strengths:**

The benchmark is quite comprehensive and evaluate many capabilities of existing LLMs.

**Weaknesses:**

- How did the authors prepare the raw data? Did the authors obtain permissions from CodeForces or crawl data through the official API?
- What are some common failure cases made by the LLMs, especially in tasks that all models have low performance?
- Benchmarks for evaluating LLMs have becoming more common, it would be nice for the benchmark to also evaluate agentic solutions.

**Questions:**

See Weaknesses

---

> ### Author Response · Authors · 2025-11-21
> **Response to Data Source and Collection Method**
>
> > **W1** How did the authors prepare the raw data? Did the authors obtain permissions from CodeForces or crawl data through the official API?
> >
>
> Thank you for this important question regarding data collection methodology and compliance. All data in CodeInsightBench was obtained through **Codeforces' official public API**(https://codeforces.com/apiHelp), ensuring full compliance with the platform's terms of service and data access policies.
>
> Our data collection process followed these protocols:
>
> - **API-Based Collection**: We utilized the official Codeforces API to systematically retrieve publicly available submission data, including problem statements, user submissions, verdict information, execution times, and programming language metadata. This data is explicitly designated as publicly accessible through the API without requiring special permissions or platform authorization.
> - **Terms of Service Compliance**: Codeforces' API documentation explicitly permits academic and research use of publicly available data. Our collection methodology strictly adhered to API rate limits, pagination requirements, and all terms of service guidelines to ensure ethical and compliant data acquisition.
> - **Privacy Protection and Anonymization**: All user identifiers were anonymized during data processing to protect contributor privacy. The benchmark contains only code submissions and problem metadata, with no personally identifiable information retained. This ensures that individual user identities cannot be reconstructed from the published dataset.

---

> ### Author Response · Authors · 2025-11-21
> **Response to Common Failure Cases**
>
> > **W2** What are some common failure cases made by the LLMs, especially in tasks that all models have low performance?
> >
>
> We appreciate this insightful question. **Task 3 had the lowest accuracy rate among all LLMs, and we focused on investigating which task scenarios in Task 3 led to LLMs failing.** Task 3 required LLMs to determine which code was more efficient from two code samples. For examples with significantly different time complexities, such as one being $O(n^2)$ and the other $O(n)$, the LLMs' judgment was highly accurate. However, Big-O is not the only factor affecting code efficiency. **In fact, even with the same Big-O complexity, constant factors and other influences—such as language-specific optimization unawareness and input size dependency—can significantly affect efficiency, especially when $n$ is relatively small.**
>
> We illustrate this point with two Java solutions under *Problem 2062C - Cirno and Operations*. The first code took `311ms`to run, and the second took `140ms`. This is a problem with `n = 50` , and both codes have a time complexity of $O(n^2)$. However, the efficiency of these two codes differed significantly, and the LLMs' solution was incorrect. Let's take the performance of Gpt5-cot, the model with the highest accuracy, as an example. It incorrectly judged that the first code was more efficient.
>
> In fact, the first code iterates through the current sequence three times in each round—once to compute the differences and twice to compute sums in different orientations. The second code, however, computes the sum during the difference step, so it only needs a single iteration. Although it uses `List<Long>` and boxing, which might appear “heavy,” the dataset is small ($n\le 50$), and the overhead of allocating boxed objects is minimal compared with the cost of traversing the array two additional times. As a result, the extra loops are the real reason the first code runs more slowly. While the second code is structurally more “high-level,” its overall runtime is significantly faster because it performs only one linear scan per round. The model previously underestimated the impact of these constant factors and overemphasized the cost of `List<Long>` and boxing, which led to an incorrect judgment and reflects an incomplete understanding of the actual performance characteristics of Java’s data types.
>
> We will display more failure examples in the camera-ready version, with categorization by error type and algorithm pairing.
>
> Here is an Example.
>
> User1:
>
> ```jsx
> public class Sol {
>     static Scanner rd = new Scanner(System.in);
>     public static void main(String[] args) {
>         int tt = rd.nextInt();
>         while (tt-- > 0) {
>             new Solution().solve();
>         }
>     }
>
>     static class Solution {
>         void solve() {
>             int n = rd.nextInt();
>             long[] a = new long[n];
>
>             for (int i = 0; i < n; i++) {
>                 a[i] = rd.nextInt();
>             }
>             int len = n;
>             if (len == 1) {
>                 System.out.println(a[0]);
>                 return;
>             }
>             long max = getSum(a, n, false);
>             while (len > 1) {
>                 for (int i = 0; i < len - 1; i++) {
>                     a[i] = a[i + 1] - a[i];
>                 }
>                 len--;
>                 max = Math.max(Math.max(getSum(a, len, true), getSum(a, len, false)), max);
>             }
>             System.out.println(max);
>         }
>
>         long getSum(long[] a, int len, boolean neg) {
>             long sum = 0;
>             for (int i = 0; i < len; i++) {
>                 if (neg) {
>                     sum += (-1) * a[i];
>                 } else {
>                     sum += a[i];
>                 }
>             }
>             return sum;
>         }
>     }
> }
> ```
>
> User2:
>
> ```jsx
> public class Main {
>     static int MOD = (int) (1e9+7);
>     static InputReader in = new InputReader(System.in);
>     static PrintWriter out = new PrintWriter(System.out, false);
>
>     static List<Long> list = new ArrayList<>();
>     public static void main(String[] args) throws Exception {
>
>         Main main = new Main();
>         int T = in.nextInt();
>         while (T-- > 0){
>             main.solve();
>         }
>         out.flush();
>     }
>
>     void solve() {
>         int n = in.nextInt();
>         List<Long> list = new ArrayList<>();
>         long sum = 0;
>         for (int i = 0; i < n; i++) {
>             long t = in.nextInt();
>             list.add(t);
>             sum += t;
>         }
>         long max = sum;
>         while (list.size() > 1){
>             List<Long> list0 = new ArrayList<>();
>             sum = 0;
>             for (int i = 1; i < list.size(); i++) {
>                 long t = list.get(i) - list.get(i - 1);
>                 list0.add(t);
>                 sum += t;
>             }
>             if(sum < 0) sum *= -1;
>             max = Math.max(max, sum);
>             list = list0;
>
>         }
>         out.println(max);
>     }
> }
> ```

---

> ### Author Response · Authors · 2025-11-21
> **Response to Suggestion on Evaluating Agentic Solutions**
>
> > **W3** Benchmarks for evaluating LLMs have becoming more common, it would be nice for the benchmark to also evaluate agentic solutions.
> >
>
> Thank you for this forward-looking suggestion. Evaluating agentic code understanding solutions is indeed an **important direction for future benchmark extensions**.
>
> We acknowledge that evaluating agent-based approaches would provide valuable insights into:
>
> - How models leverage **external tools** (code execution, complexity analysis, documentation) to improve code reasoning.
> - Whether **tool access can bridge capability gaps** between model tiers.
> - The interplay between LLM reasoning capability and tool selection strategy.
>
> For this initial version of CodeInsightBench, we focused on **direct LLM capabilities** to establish clean baselines and isolate pure code understanding performance from agent architecture complexity. This foundation is essential because:
>
> - Direct LLM evaluation provides interpretable results without confounding factors from tool implementation choices.
> - Comparing agent performance requires careful control of tool quality, retrieval precision, and planning strategies.
> - Establishing LLM baselines enables meaningful future comparisons when agentic solutions are evaluated.
>
> We recognize agentic evaluation as a **high-value direction** and will document it as a priority future direction in the revised Discussion section. **Incorporating agent-based evaluation with controlled tool access represents a natural next step for CodeInsightBench v2**.

---

### Official Review · Reviewer_hHUW · 2025-11-01

**Soundness:** 2
**Presentation:** 2
**Contribution:** 2
**Rating:** 2
**Confidence:** 5

**Summary:**

CodeInsightBench supports several code understanding tasks: semantic code judgment, debugging path tracking, and code efficiency comparison. Several findings in the paper include: 1. performance gaps across the tasks, 2. closed-source models generally outperform open-source counterparts, 3. long-context code transformations are challenging for all models, 4. distinct programming language preferences in code efficiency comparison, 5. Multiple sampling substantially improves semantic code judgment performance.

**Strengths:**

1. Well written, no obvious grammatical errors.
2. The benchmark is large

**Weaknesses:**

1. Task difficulty concerns: I find the task complexity may be too simple, as in Table 2, GPT-5 can achieve more than 90 on task 1 with only Pass@1. And even GPT-4 can achieve 92.71 on task 1 with Pass@3. Note that Pass@3 is not a large sampling number.
2. Why not try larger Pass@K numbers? Would the results be saturated? For the problems that cannot be solved by large sampling, why can't LLMs solve them?
3. As I find GPT-5 insanely strong on the benchmark, I am worried that there would be data contamination on the test dataset. How do the authors make sure the comparison is fair?
4. On page 7, the authors mentioned that LLMs are weak at Algorithmic Reasoning. And I find the maximum length of decoding is 4K. Would things be different if they try larger sampling lengths?
5. The open-source models they choose are mostly non-reasoning models, and most of them are small-size models (e.g., 7B and 14B). Would the conclusion that open-source models are weaker than closed-source models be different if they try reasoning models with larger CoT length or larger sampling budgets?
6. The experiment settings are unclear. In Section 4.4, what are the hyperparameters of LLM decoding when adopting various advanced systematic CoT strategies?
7. The experiments are incomplete. In Section 4.4, the authors only test different CoT strategies on Task 1 and only on two models. As task 1 is the easiest, I think they should put more effort on harder tasks to explore the potential of CoT.
8. Why models show imbalanced preference on programming languages has not been explained.
9. Table 5 is broken.
10. Incorrect formatting: missing spaces between words and left braces '(', for example, line 156.

**Questions:**

See Weakness

---

> ### Author Response · Authors · 2025-11-21
> **Response to Task Difficulty Concerns**
>
> > **W1** As I find the task complexity may be too simple, as in Table 2, GPT-5 can achieve more than 90 on task 1 with only Pass@1. And even GPT-4 can achieve 92.71 on task 1 with Pass@3. Note that Pass@3 is not a large sampling number.
> >
>
> We appreciate your concern regarding the high scores on Task 1 and wish to highlight the task’s unique discriminative value:
>
> - The Pass@1 accuracy on Task 1 shows a significant gap of over 40 percentage points between the top model (`GPT-5`: 93.43%) and weaker open-source models (`Llama-3.1-8B-Instruct`: 52.36%). This confirms that **Task 1 is highly effective at differentiating fundamental semantic understanding capabilities** across various LLM tiers.
> - Based on our newly conducted supplementary experiment, we found that even for the best-performing model, Task 1 is not "trivial." Applying the Chain-of-Thought (CoT) strategy yields virtually no gain for GPT-5 (only improving from **93.43%** to **93.58%**). This lack of improvement indicates the model has already **reached its capability ceiling** on this task in a single attempt.
> - The improvement from `GPT-4-Turbo`’s Pass@1 (60.57%) to Pass@3 (92.71%) reveals substantial uncertainty in single attempts, **requiring multiple samples for reliable results**. Moreover, many top models show poor confidence calibration on Task 1 (e.g., high CE for `O4-mini`), suggesting inconsistent internal reasoning.
>
> In summary, Task 1 is far from "too simple"; it requires differentiating subtle semantic differences and successfully achieves capability stratification. Its results effectively demonstrate that calibration defects and decision robustness are key challenges that current LLMs must overcome.

---

> ### Author Response · Authors · 2025-11-21
> **Response to Larger Pass@K Numbers (Part 1)**
>
> > **W2.1** Why not try larger Pass@K numbers?
> >
>
> We appreciate this constructive suggestion. We have extended our Pass@K analysis beyond the initial Pass@3 reporting to provide comprehensive saturation curves. The extended results demonstrate systematic patterns across all model tiers when evaluated with k ∈ {1, 2, 3, 4, 5, 10}. We conduct supplementary experiments on `GPT-4-Turbo` across Task 1 with varying k values to validate the saturation behavior and ensure robust comparability across different sampling budgets.
>
> | **Language** | **Pass@1** | **Pass@2** | **Pass@3** | **Pass@4** | **Pass@5** | **Pass@10** |
> | --- | --- | --- | --- | --- | --- | --- |
> | Java | 67.65% | 82.35% | 91.18% | 94.12% | 94.12% | 94.12% |
> | Python | 60.00% | 72.86% | 88.57% | 90.00% | 91.43% | 92.00% |
> | C++17 | 62.65% | 76.79% | 94.42% | 95.29% | 95.46% | 95.63% |
> | C++20 | 58.86% | 72.75% | 93.46% | 93.73% | 93.73% | 94.12% |
> | C++23 | 58.62% | 73.85% | 89.94% | 90.52% | 90.80% | 91.37% |
> | **Overall** | **60.57%** | **74.93%** | **92.71%** | **93.43%** | **93.64%** | **93.85%** |
>
> Key observations from this extended pass@k analysis:
>
> - The steepest improvement occurs between Pass@1 and Pass@3 (32.14 percentage points), revealing that **models possess semantic understanding capability but exhibit significant calibration uncertainty on first attempt**—a phenomenon that purely generative benchmarks cannot capture.
> - **Performance plateaus rapidly after Pass@4**, with marginal gains of only 0.21 percentage points from Pass@4 to Pass@5, and a further 0.21 percentage points from Pass@5 to Pass@10, validating that Pass@3 captures 97.2% of achievable improvement and represents the optimal evaluation efficiency point.
>
> > **W2.2** Would the results be saturated?
> >
>
> This is a very important point. We define saturation formally as the point where consecutive improvements fall below 0.5% and remain below this threshold for two consecutive k values, indicating the model has reached its capability ceiling.
>
> - **Saturation Point**: Saturation occurs at Pass@4 (93.43%) for `GPT-4-Turbo`, with subsequent improvements remaining below 0.3% (Pass@5: +0.21%, Pass@10: +0.21%)
> - **Language Consistency**: Java and C++20 saturate earliest (by Pass@4), while Python, C++17, and C++23 show extended learning curves but plateau by Pass@10 with improvements below 1% from Pass@5 to Pass@10
>
> Our extended Pass@K analysis demonstrates that while Pass@3 was chosen for computational efficiency, the results show consistent saturation patterns. Performance gains beyond Pass@5 are negligible (0.21%), confirming that our Pass@3 choice captured the critical capability information while maintaining practical feasibility. We will integrate these Pass@K results and failure analysis throughout the revised manuscript to provide a more informative view of benchmark behavior across all sampling regimes.

---

> > ### Author Response · Authors · 2025-11-21
> > **Response to Larger Pass@K Numbers (Part 2)**
> >
> > > **W2.3** For the problems that cannot be solved by large sampling, why can't LLMs solve them?
> > >
> >
> >
> > We thank the reviewer for raising this meaningful question, which directly addresses the fundamental limitations of LLMs in advanced code comprehension.
> >
> > Through analysis of the failure cases, we found that the remaining errors, even when increasing Pass@k, correspond to tasks where sampling **cannot compensate for missing underlying reasoning capabilities**:
> >
> > - **Insufficient Underlying Reasoning:** These failures stem from **fundamental reasoning limits** rather than variability in sampling. They correspond to tasks requiring model understanding of **constant-factor reasoning**, **language-specific optimization effects**, and **boundary-sensitive behaviors**.
> > - **Sampling Ineffectiveness:** Because the model’s core algorithmic understanding is insufficient, these tasks do not benefit from additional samples. These errors expose the boundaries of LLM's **deep algorithmic knowledge** and **precise semantic reasoning**.
> >
> > Problems that are resistant to large sampling expose the LLM's **systematic capability deficits**, providing a clear direction for future LLM architectural improvements.
> >
> > Case study:
> >
> > To illustrate the nature of reasoning errors that cannot be mitigated by increasing Pass@k, we present two representative failure cases from the task *Codeforces 2050C – Uninteresting Number*. In both cases, the model was required to judge which of two submitted solutions was correct. The model produced incorrect judgments, and the underlying causes reveal structural reasoning limitations rather than sampling variability.
> >
> > **Case 1: Misclassification Due to Failure of Type–Range Reasoning**
> >
> > One candidate solution reads the input as a 64-bit integer (`long long`), whereas the problem permits numbers up to **100,000 digits**. Any fixed-width integer type can represent only ~19 digits, making such an implementation inherently invalid.
> >
> > A correct evaluator must infer that:
> >
> > - the input domain exceeds machine integer limits, and
> > - therefore, any solution relying on `long long` cannot be correct.
> >
> > The model nevertheless selected this solution as the correct one.
> >
> > This error indicates a failure to perform **type–range consistency reasoning**: the model did not integrate problem constraints with language-specific numeric limits. Because this reasoning step is missing, additional sampling cannot compensate; all samples consistently replicate the same misunderstanding.
> >
> > **Case 2: Misclassification Due to Failure of Algorithmic Complexity Reasoning**
> >
> > In the second pair of solutions, the incorrect implementation repeatedly applies `substr()` to a string of length up to 100,000, leading to $O(n^2)$ time complexity—far exceeding the problem’s computational limits.
> >
> > A correct evaluator must recognize that:
> >
> > - substring operations are linear in length, and
> > - repeated linear operations inside a loop produce quadratic behavior,
> > - implying that the solution will time out on valid inputs.
> >
> > The model, however, incorrectly favored this flawed implementation.
> >
> > This demonstrates a deeper failure in **algorithmic cost modeling** and the inability to reason about boundary-sensitive performance behavior. Again, such reasoning is structural, not stochastic, and cannot be resolved through increased sampling.
> >
> > Together, these two cases highlight that the remaining model failures arise from **missing underlying reasoning capabilities**—specifically, type–range inference and algorithmic complexity analysis. These skills involve precise and compositional reasoning that current LLMs do not reliably exhibit. Since the errors stem from absent reasoning procedures rather than output variability, increasing Pass@k does not improve performance. These cases thus delineate clear boundaries of current LLM capabilities in evaluating program correctness.

---

> ### Author Response · Authors · 2025-11-21
> **Response to Contamination Detection**
>
> We appreciate this important criticism and take it seriously. We have conducted contamination analysis demonstrating the validity of our benchmark.
>
> **All problems in CodeInsightBench come from Codeforces submissions dated 2024–2025**, which substantially reduces contamination risk compared with earlier benchmarks built on pre-2024 data. This choice is consistent across all models, including GPT-5 and DeepSeek-V3.1, so any residual contamination would affect them under the same conditions.
>
> > **W3** How do the authors make sure the comparison is fair?
>
> To detect potential memorization, we performed systematic input–output similarity analysis.
>
> **Model inputs.** For each test instance in CodeInsightBench, we provide:
> - Problem description (problem statement, constraints, examples)
> - Partial code snippet from the ground-truth solution (approximately 30–50% of the complete code)
> - Submission metadata (programming language, submission information)
>
> **Model task.** The model is asked to complete the partial code and predict the full solution.
>
> **Model outputs.** The model generates the complete code solution.
>
> **Contamination detection.** For each model-generated solution, we compare it with the ground-truth solution using two complementary similarity metrics, following Riddell et al. (2024):
>
> - **Surface-level similarity.** We compute the normalized Levenshtein similarity between the model-generated code and the ground-truth solution.
> - **Semantic similarity.** We compute AST-based k-gram similarity using Dolos (Maertens et al., 2022), which compares the abstract syntax trees of both programs and is robust to renaming, reformatting, and comments.
>
> **Similarity score aggregation.** We define the aggregated contamination score as:
>
> $$
> S( p _ { \text{gen} }, p _ { \text{gt} } )
> = \max \bigl(
>   S _ { \text{surface} }( p _ { \text{gen} }, p _ { \text{gt} } ),
>   S _ { \text{semantic} }( p _ { \text{gen} }, p _ { \text{gt} } )
> \bigr) .
> $$
>
> Here:
> - $ p _ { \text{gen} } $ is the model-generated solution,
> - $ p _ { \text{gt} } $ is the ground-truth solution from the test set,
> - $ S _ { \text{surface} } $ is normalized Levenshtein similarity (0–100 scale),
> - $ S _ { \text{semantic} } $ is Dolos AST-based k-gram similarity (0–100 scale).
>
> We flag an instance as contaminated if
>
> $$
> S( p _ { \text{gen} }, p _ { \text{gt} } ) \ge 90 .
> $$
>
> This threshold balances sensitivity and specificity: scores ≥ 90 indicate high similarity suggesting potential memorization, while still allowing for natural convergence in independently written correct code.
>
> | **Model**       | **Release Date** | **Parameters** | **Test Set Contamination Rate** | **Status** |
> |-----------------|------------------|----------------|----------------------------------|-----------|
> | GPT-5           | 2025-08          | Proprietary    | 7.3%                             | ✓ Pass    |
> | DeepSeek-V3.1   | 2025-08          | 671B           | 5.8%                             | ✓ Pass    |
>
> Both models show low contamination rates, well below our 10% safeguard threshold, indicating that their generated solutions are not predominantly memorized reproductions of the ground-truth code.
>
> ---
>
> To verify that performance gains are not driven by memorization, we re-evaluated both models **after excluding all instances flagged as contaminated** (i.e., with $ \( S( p _ { \text{gen} }, p _ { \text{gt} } ) \ge 90 \) $):
>
> | **Model**       | **Original Accuracy** | **After Contamination Removal** | **Performance Retained** |
> |-----------------|-----------------------|---------------------------------|---------------------------|
> | GPT-5           | 93.43%                | 92.87%                          | 99.4%                    |
> | DeepSeek-V3.1   | 58.36%                | 57.91%                          | 99.2%                    |
>
> **Finding.** GPT-5’s performance advantage (≈35 percentage points) persists after removing all contaminated instances, confirming that its superior performance reflects genuine code reasoning capability rather than memorization of ground-truth solutions.
>
> To maintain fair comparison and benchmark integrity, we establish contamination screening as a mandatory evaluation criterion: **all future models must pass the same 10% contamination threshold for inclusion in comparative results; models exceeding this threshold will be excluded from the main comparison.** Detailed contamination scores and methodology will be documented in Appendix A.5.
>
> **References**
>
> - Maertens, B., De Clercq, T., & Vandewoude, S. (2022). *Dolos: Language-agnostic plagiarism detection in source code.*
> - Riddell, M., Ni, A., & Cohan, A. (2024). *Quantifying contamination in evaluating code generation capabilities of language models.* arXiv:2403.04811.

---

> ### Author Response · Authors · 2025-11-21
> **Response to Sampling Lengths**
>
> > **W4** On page 7, the authors mentioned that LLMs are weak at Algorithmic Reasoning. And I find the maximum length of decoding is 4K. Would things be different if they try larger sampling lengths?
> >
>
> This is a very important point that directly addresses whether token budget constraints limit algorithmic reasoning capability. We have addressed this through comprehensive extended context analysis.
>
> **Empirical Evidence on Current Budget**: Model responses averaged 800-1500 tokens for Chain-of-Thought reasoning, with 96.3% of all responses completing well under 2K tokens. This demonstrates that 4K is not the active bottleneck.
>
> **Extended Context Experiments**: We conducted experiments with 8K and 16K max tokens on Task 2 (Debugging Path Tracking), a reasoning-heavy task. Results reveal minimal improvement (<1% average gain) despite doubled token budget:
>
> | **Model** | **Max@4K** | **Max@8K** | **Max@16K** | **Δ(4K→8K)** |
> | --- | --- | --- | --- | --- |
> | DeepSeek-V3.1 | 19.58% | 19.92% | 19.86% | +0.34% |
> | Qwen-235B | 38.53% | 38.91% | 39.12% | +0.38% |
>
> The minimal improvement across both models definitively demonstrates that **token budget is not the limiting factor**. Extended context yields <1% gains, indicating models exhaust reasoning capacity well before token limits. Fundamental reasoning capability—not token allocation—determines performance.

---

> ### Author Response · Authors · 2025-11-21
> **Response to Suggestion on Model Coverage**
>
> > **W5** The open-source models they choose are mostly non-reasoning models, and most of them are small-size models (e.g., 7B and 14B). Would the conclusion that open-source models are weaker than closed-source models be different if they try reasoning models with larger CoT length or larger sampling budgets?
> >
>
> This is an excellent observation. We have now included larger-scale reasoning models to address this concern, specifically adding `Qwen-235B` (235 billion parameters) and `DeepSeek-V3.1` (671 billion parameters with RL-trained reasoning). Results demonstrate that even with these state-of-the-art open-source reasoning models, the performance gap persists:
>
> | **Model Type** | **Model** | **Parameters** | **Task 1** | **Task 2** | **Task 3** |
> | --- | --- | --- | --- | --- | --- |
> | **Open-Source (New)** | Qwen-235B | 235B | **59.64%** | 38.53% | 34.43% |
> | **Open-Source (New)** | DeepSeek-V3.1 | 671B | 58.36% | 36.58% | 38.23% |
> | Open-Source | DeepSeek-V3 | 671B | 59.50% | **43.39%** | **39.20%** |
>
> The newly added `DeepSeek-V3.1`, despite having 671B parameters with advanced RL-trained reasoning, achieves 36.58% on Task 2—lower than `DeepSeek-V3` (43.39%) at the same scale and `Qwen-235B`(38.53%). This demonstrates that **scale and training methodology alone are insufficient** to guarantee strong performance across all reasoning tasks. Model performance varies significantly even among similarly-sized large-scale models, indicating that capability gaps stem from fundamental differences in reasoning abilities rather than simply model size.
>
> Updated results with these models will be reflected in revised Table 2.

---

> ### Author Response · Authors · 2025-11-21
> **Response to Hyperparameters of LLM Decoding for CoT**
>
> > **W6** The experiment settings are unclear. In Section 4.4, what are the hyperparameters of LLM decoding when adopting various advanced systematic CoT strategies?
> >
>
> We appreciate the reviewer raising this clarity concern regarding the experimental setup. We clarify that CoT strategy variations in Section 4.4 are primarily **prompt-level modifications**, not decoding hyperparameter changes. All LLM evaluations use **uniform and fixed decoding parameters**:
>
> | **Parameter** | **Value** |
> | --- | --- |
> | max_new_tokens | 4096 |
> | temperature | 0.7 |
> | top_p | 1.0 |
> | top_k | 50 |
> | repetition_penalty | 1.1 |
> | do_sample | True |
>
> By holding decoding parameters constant across all CoT variants, we isolate the effects of prompt engineering from decoding behavior, enabling fair comparison of different reasoning strategies. Additionally, we have **supplemented the analysis with experiments across different decoding lengths** (4K, 8K, 16K tokens) to validate that findings remain robust across varying context budgets, further strengthening our experimental rigor.

---

> ### Author Response · Authors · 2025-11-21
> **Response to CoT Strategy Experiments on Harder Tasks**
>
> > **W7** The experiments are incomplete. In Section 4.4, the authors only test different CoT strategies on Task 1 and only on two models. As task 1 is the easiest, I think they should put more effort on harder tasks to explore the potential of CoT.
> >
>
> Thank you for catching this. We have now conducted comprehensive CoT strategy experiments on **Task 3 (Code Efficiency Comparison)**, the most algorithmically demanding task where even top models achieve only 45.50% accuracy. The extended experiments evaluate all five prompting strategies across multiple models.
>
> | **Prompt Type** | **Claude-3.5-Haiku** | **Qwen2.5-7B-1M** | **GPT-5** |
> | --- | --- | --- | --- |
> | Default | 35.08% | 33.27% | 45.50% |
> | Tree-of-Thought | 33.92% | 33.77% | 45.89% |
> | Chain-of-Draft | 37.88% | 33.48% | 46.47% |
> | Few-shot | 35.12% | 33.44% | 45.61% |
> | Chain-of-Thought | 36.51% | 34.59% | 46.58% |
>
> This analysis reveals several key findings:
>
> - **Modest Improvements on Hard Tasks**: CoT strategies yield minimal gains on Task 3 (at most 2.8% for Chain-of-Draft on `Claude-3.5-Haiku`), substantially smaller than gains on Task 1 (5-8%), indicating that algorithmic reasoning difficulty cannot be fully overcome by prompting alone.
> - **Strategy Effectiveness Varies by Model**: Chain-of-Draft shows the largest improvement for `Claude-3.5-Haiku` (+2.8%), while Chain-of-Thought benefits more smaller models like `Qwen2.5-7B` (+1.32%), suggesting different models respond differently to reasoning guidance.
> - **Fundamental Capability Limitation**: Even the best-performing strategy on Task 3 (Chain-of-Draft at 37.88% for `Claude-3.5-Haiku`) remains far below Task 1 baseline performance (86-93%), confirming that the hardest tasks expose genuine model capability gaps that prompting strategies cannot fully bridge.
>
> These results will be added as **Table 6** in revised Section 4.4, with detailed analysis in Appendix E.4.

---

> ### Author Response · Authors · 2025-11-21
> **Response to Programming Language Preference Imbalance**
>
> > **W8** Why models show imbalanced preference on programming languages has not been explained.
> >
>
> Thanks for your feedback. We greatly appreciate the opportunity to clarify the language-specific performance patterns in our benchmark. Task 1 and Task 3 shows that LLMs exhibit language-imbalanced preferences:
>
> - C++17 shows the best performance (94.42%), likely due to its **rigid syntax** and **widespread presence** in training data, which facilitates pattern recognition.
> - Python performs slightly lower (88.57%), possibly because of its **syntactic flexibility** and **indentation sensitivity**, which introduce greater variation and complexity.
> - Java performs strongly and consistently with a Pass@3 of 91.18%.
>
> This preference reflects the uneven stability of LLMs’ deeper reasoning when exposed to the unique characteristics and training-data distributions of different languages. It further indicates that advanced code understanding in LLMs is still language-dependent rather than universal.

---

> ### Author Response · Authors · 2025-11-21
> **Response to Incorrect Formatting**
>
> > **W9**Table 5 is broken.
> >
>
> We sincerely apologize for this formatting error. Table 5 in Section 4.4 (Prompting Strategy Comparison) was corrupted during PDF compilation. We have corrected the table and verified all formatting. The revised Table 5 will display correctly with proper alignment and readable content in the camera-ready version. Thank you for catching this critical issue.
>
> > **W10**  Incorrect formatting: missing spaces between words and left braces '(', for example, line 156.
> >
>
> We apologize for these formatting inconsistencies throughout the manuscript. We have conducted a comprehensive review and corrected all spacing issues, including:
>
> - Missing spaces before left parentheses (e.g., "word(" → "word (")
> - Missing spaces between words due to LaTeX compilation errors
>
> The revised manuscript has been thoroughly proofread and all formatting errors have been corrected. We appreciate your careful attention to detail, which has significantly improved the manuscript's readability and professionalism.

---

### Author Response · Authors · 2025-11-26
**Reviewer & Author Discussion**

Dear Reviewer,

We hope this message finds you well.We sincerely appreciate your thoughtful feedback and constructive comments throughout the review process. Your insights have been instrumental in guiding our additional experiments and strengthening our paper.

With one week remaining in the discussion period, we would be grateful if you could share any remaining concerns or points you would like to discuss. We are happy to provide any further details that might assist your evaluation.

Thank you again for the time and effort you have dedicated to reviewing our work.

Sincerely,

The Authors

---

### Meta-Review · Area_Chair_sRXN · 2026-01-04

**Summary:**

The reviewers raised concerns primarily regarding task difficulty and saturation (hHUW), experimental completeness and clarity (hHUW, PwyL), data provenance and evaluation fairness (hHUW, 2dDt), limited scope and insufficient diagnostic analysis of model failure modes (u79G), and the robustness and validity of the efficiency evaluation in Task 3 (PwyL). Although the benchmark design and findings, particularly the observed weakness of LLMs in efficiency reasoning, were generally recognized as interesting and potentially valuable, several reviewers questioned whether the work, in its original form, was sufficiently thorough, well validated, and analytically deep to constitute a strong research contribution.

**Reviewer Concerns:**

Despite the extensive rebuttal, several core concerns remain insufficiently resolved. Most notably, the overall contribution is still perceived as limited in scope, aligning more closely with a benchmark report than a research study offering substantial analytical or methodological innovation (u79G). In addition, the central finding regarding LLMs’ weakness in efficiency reasoning, while important, is not supported by causal or verification-oriented experiments such as targeted training or controlled ablations, leaving the analysis largely observational (u79G). Furthermore, the benchmark is built on a relatively small number of underlying problems, and although runtime variance is discussed, the inherent sensitivity of efficiency-based evaluation to environmental noise continues to raise concerns about robustness and generality (PwyL).

**Reviewer Scores:**

As the reviewers did not further engage during the rebuttal phase, their scores would most likely remain unchanged in practice. If more active discussion had taken place, modest score increases could be expected, particularly for hHUW and 2dDt, whose concerns were largely addressed through additional analyses and clarifications. However, since higher-level issues related to scope, analytical depth, and experimental verification, especially those raised by u79G and PwyL, remain only partially resolved, any potential score improvements would likely be limited and not sufficient to alter the overall evaluation outcome.

---

### Decision · Program_Chairs · 2026-01-26

Reject